# Position: Current Benchmarking Hinders Real Progress in Deep Learning for Time Series Forecasting

**Valentina Moretti** [1]   **Ivan Marisca** [1]   **Cesare Alippi** [1,2]   **Andrea Cini** [3]

## Abstract

Deep learning models have grown popular in time series applications. However, the large quantity of newly proposed architectures and the often contradictory empirical results make it difficult to assess which design choice and model component drives performance. In this position paper, we argue that current benchmarking practices fail to identify the factors responsible for performance differences, thus slowing down progress in the field. In particular, differences in crucial design dimensions are overlooked when comparing architectures, ultimately leading to inconsistent outcomes. To support our position, we show that such differences–often treated as mere implementation details–can have a greater impact than adopting specific sequence modeling layers. We discuss how overlooked aspects (such as globality and locality) can (1) fundamentally change the class of the forecasting method and (2) drastically affect empirical results. Our findings suggest rethinking our benchmarking practices and focusing on the foundational aspects of the forecasting problem when designing and comparing architectures. As a concrete step, we propose an *auxiliary forecasting model card*, i.e., a template with a set of fields to characterize existing and new forecasting architectures based on key design choices.

## 1. Introduction

Novel sequence modeling architectures are consistently improving the state of the art (SOTA) in many applications, such as in natural language processing (Gu et al., 2022; Gu and Dao, 2024; Beck et al., 2024). In contrast, results in time series forecasting–despite advances in foundation models (Das et al., 2024; Ansari et al., 2025)–offer a much more uncertain way ahead. Recent work questions the actual effectiveness of modern deep learning architectures in favor, e.g., of simpler models (Toner and Darlow, 2024; Zeng et al., 2023). Indeed, **current research is seemingly stuck in a loop of positive results being quickly dismissed by new evidence that reveals gaps in our understanding of the components that contribute to accurate forecasts** (Shao et al., 2024; Tan et al., 2024; Brigato et al., 2026). As a consequence, the community is striving to address these issues and find explanations for them. Brigato et al. (2026), for example, shows that inconsistent hyperparameter tuning and bias in dataset selection can mislead, and that no architecture significantly outperforms the others under a fair comparison. Other works, instead, have focused on improving the quality and diversity of available benchmarks (Qiu et al., 2024). Although these aspects are part of the problem, we argue that there are **conceptual issues in current benchmarking practices beyond common pitfalls in empirical evaluation**. Position: **We believe that our current approach to building and comparing deep learning architectures for time series forecasting overlooks fundamental design dimensions and their interplay. As a result, it fails in explaining observed empirical results and supporting progress in the field.**

**Just implementation details?**   Many recent forecasting architectures stack and combine different components and operators, introducing many–often hidden–implementation choices (Zhou et al., 2021; Wu et al., 2021; Liu et al., 2022a; Nie et al., 2023; Liu et al., 2023; Zhang and Yan, 2023). However, the impact of such design choices on the resulting model and its performance is often not accounted for. For example, in recent work, collections of synchronous univariate time series are often treated as single multivariate signals. Although this might sound like a minor formalization aspect, it results in misconceptions and results that are difficult to interpret. Indeed, as a result, multivariate models are often compared against global univariate forecasting architectures (Montero-Manso and Hyndman, 2021) in settings where the latter clearly have an advantage due to sample efficiency and the curse of dimensionality (global models share

[1]IDSIA, Università della Svizzera italiana, Lugano, Switzerland [2]Politecnico di Milano, Milan, Italy [3]IMOS Lab, EPFL, Lausanne, Switzerland. Correspondence to: Valentina Moretti <valentina.moretti@usi.ch>, Andrea Cini <andrea.cini@epfl.ch>.

*Proceedings of the 43rd International Conference on Machine Learning*, Seoul, South Korea. PMLR 306, 2026. Copyright 2026 by the author(s).

parameters among time series). This has led to attributing performance gains to architectural differences, e.g., in sequence modeling operators, rather than to well-understood principles for forecasting groups of time series (Salinas et al., 2020; Montero-Manso and Hyndman, 2021). This is only an example of problems that involve many of the design dimensions inherent in designing forecasting architectures. These include, e.g., methods for modeling dependencies across variates and related time series (Liu et al., 2023; Zhang and Yan, 2023), the use of exogenous inputs, and more.

**Are benchmarks measuring actual progress?** Two of the main objectives of good benchmarking are (1) assessing which design works best in different scenarios, and (2) ensuring progress in the field. As we will show, comparing the aforementioned complex architectures against similarly complex SOTA models fails at objective (1) as it cannot attribute performance gains to specific components. Moreover, overlooking design dimensions specific to the time series forecasting problem, as we discuss, hinders (2) as well, as observed performance gains may simply stem from failure in factoring out specific differences in the design and implementation of the baselines being compared, rather than from methodological improvements. We argue that, to enable meaningful performance comparisons, benchmarks should disentangle the effects on performance of distinct design dimensions. Moreover, to avoid misleading conclusions from reported results, key forecasting design choices should be made explicit.

**An analysis of current failure points** To pinpoint the specific sources of inconsistencies often observed in benchmarking results, we focus on four key design dimensions that significantly affect model performance: *D1. model configuration*, i.e., selecting among different approaches to forecasting multiple time series (e.g., local, global, or hybrid); *D2. preprocessing and exogenous variables*, i.e., selecting exogenous variables and setting up preprocessing and postprocessing operations; *D3. temporal processing*, i.e., accounting for temporal (intra-series) dependencies; *D4. spatial processing*, i.e., accounting for spatial (inter-series) dependencies. While some of these dimensions are not always orthogonal (e.g., space and time can be processed in an integrated fashion), we argue that analyzing how these aspects affect recent results is key to understanding the shortcomings of current benchmarking practices. We take the following steps to support our position:

- We analyze the current state of deep learning for time series forecasting by relying on principles to forecast groups of time series to explain the often contradictory empirical results.

- We empirically quantify the impact of overlooked design choices and implementation details in SOTA archi-

tectures, and show that they explain a significant portion of the observed performance improvements.

- We show that a streamlined architecture built on well-understood design principles can match the performance of the current SOTA.

- We introduce an *auxiliary forecasting model card* template[1]–complementary to existing generic model cards (Mitchell et al., 2019)–aimed at supporting model designers and practitioners to characterize and understand existing and new forecasting architectures.

**Our position does not dismiss the field's progress–which is tangible in many applications–but aims to advance it by fostering awareness of the existing flaws in our practices.** In particular, our objective is to stimulate discussion on our approach–as a community–to conducting machine learning research for time series forecasting. We believe that having such a discussion is an important step for the maturity of the field and to ensure future progress.

## 2. Preliminaries

### 2.1. Problem Setting

Let $\mathcal{D} = \{\boldsymbol{x}^1_{0:L_1}, \ldots, \boldsymbol{x}^N_{0:L_N}\}$ denote a collection of $N$ time series. Each $\boldsymbol{x}^i_{0:L_i} \in \mathbb{R}^{L_i \times d_x}$ is a sequence of $L_i$ observations in the interval $[0, L_i)$, where each observation has dimension $d_x$. Time series in the set can come from different domains and be generated by different stochastic processes. A binary mask, $\boldsymbol{m}^i_{0:L_i} \in \{0,1\}^{L_i \times d_x}$, may be introduced to model missing or invalid observations, due to either acquisition errors, faults, or missing channels in the case of heterogeneous time series. Exogenous variables (e.g., time encoding, calendar features, weather conditions) are denoted as $\boldsymbol{u}^i_{0:L_i} \in \mathbb{R}^{L_i \times d_u}$ and assumed to be available also when the corresponding observation is missing. If time series are synchronous, we use capital letters to denote values across the collection, e.g., $\boldsymbol{X}_t \in \mathbb{R}^{N \times d_x}$ refers to the stacked observations at time step $t$. Time series in the collection might be *correlated* (in a broad sense), i.e., uncertainty on future values of each time series might be reduced by taking into account observations from other time series.

**Forecasting groups of time series** We consider the problem of multi-step ahead time series forecasting, i.e., the problem of predicting the next $H \geq 1$ observations $\boldsymbol{x}^i_{t:t+H}$ for the i-th time series, given a window $W \geq 1$ of past observations $\boldsymbol{x}^i_{t-W:t}$ from the same time series. As the stochastic process generating data $p^i$ is unknown, the objective is to

---

[1]https://valentina-moretti.github.io/forecasting_model_cards

approximate it with a model $p_{\boldsymbol{\theta}}$ with parameters $\boldsymbol{\theta}$ such that

$$p_{\boldsymbol{\theta}}(\boldsymbol{x}_{t:t+H}^i \mid \boldsymbol{x}_{t-W:t}^i, \boldsymbol{u}_{t-W:t}^i, \boldsymbol{u}_{t:t+H}^i) \approx$$
$$p^i(\boldsymbol{x}_{t:t+H}^i \mid \boldsymbol{x}_{<t}^i, \boldsymbol{u}_{<t}^i, \boldsymbol{u}_{t:t+H}^i) \quad \forall i = 1, \ldots, N \quad (1)$$

where $\boldsymbol{x}_{<t}^i$ denotes past observations $(\boldsymbol{x}_t^i, \boldsymbol{x}_{t-1}^i, \ldots)$. We focus on the problem of obtaining *point forecasts* $\widehat{\boldsymbol{x}}_{t:t+H}^i$ of, e.g., the expected value such as $\widehat{\boldsymbol{x}}_{t:t+H}^i \approx \mathbb{E}_p\left[\boldsymbol{x}_{t:t+H}^i\right]$ by using a parametric model $\mathcal{F}(\,\cdot\,; \boldsymbol{\theta})$. Predictions are obtained by fitting parameters $\boldsymbol{\theta}$ of the chosen model family. As we will discuss in Sec. 3.1, we say that a model is *global* if its parameters are shared across all the time series. In such a case, the model is trained on the entire set of time series. Conversely, a model is *local* if its parameters are specific to a single time series. Using local models requires fitting a separate model for each series in the collection. Choosing between a local and global approach (or a hybrid thereof) depends on the task at hand, data availability, and model complexity. Global models, due to advantages in sample efficiency, are a particularly appealing choice when relying on deep learning architectures (Hewamalage et al., 2021; Benidis et al., 2022). Additionally, global models can be employed *inductively*, i.e., they can be used to forecast unseen time series, whereas local models are *transductive*. We will expand this discussion in Sec. 3.1. We provide an extended discussion of the current state of the field in App. A.

## 2.2. Evaluation Setup

Throughout the paper, we show the impact of different design choices by comparing recent SOTA architectures for long-range time series forecasting against simpler, streamlined architectures on commonly used benchmarks.

**State-of-the-art architectures** We consider representative models that shaped recent forecasting methods and that perform competitively on benchmarks. We include: 1. **PatchTST** (Nie et al., 2023), the widely used architecture that introduced "channel independence" and patch-based Transformer layers; 2. **DLinear** (Zeng et al., 2023), which combines a linear model with a time series decomposition step; 3. **TimeMixer** (Wang et al., 2024), which uses multilayer perceptrons (MLPs) to process the input at different resolutions; 4. **Linear**, a simple linear autoregressive model trained with $L2$ regularization and ordinary least squares (OLS), following Toner and Darlow (2024). We also consider models that incorporate spatial processing: 5. **iTransformer** (Liu et al., 2023), which processes the temporal dynamics with a feedforward layer and then uses standard attention among channels; 6. **ModernTCN** (Donghao and Xue, 2024), which relis on convolutional layers for spatio-temporal processing; 7. **Crossformer** (Zhang and Yan, 2023), which uses patching and spatiotemporal attention operators to model dependencies among different channels of the input time series. To ensure a fair comparison, we

evaluate all the models under the same benchmarking setup, unified settings, and with access to the same exogenous variables. We rely on the available open-source implementations of each approach and adapt them to our evaluation procedure and standardized inputs. See App. B for more details. Code available at [2].

**Reference architectures** We compare the SOTA models against reference streamlined architectures designed to evaluate the impact of different design choices along the target design dimensions. The purpose is not to propose a new architecture to challenge the SOTA. Conversely, reference architectures provide baselines, introduced to facilitate a fair and consistent comparison and to gauge the impact of different design choices more directly. The architecture stacks a temporal module and an optional spatial module. For the temporal module, we consider several alternatives: an MLP with residual connections, a temporal convolutional network (TCN) with causal dilated filters (Bai et al., 2018), a gated recurrent neural network (RNN) (Chung et al., 2014), a stack of Transformer layers (Vaswani et al., 2017), and pyramidal attention operators akin to the Pyraformer architecture (Liu et al., 2022a). In the tables, we denote these reference models as **MLP**, **TCN**, **RNN**, **Transf.**, **Pyraf.**, respectively. For the TCN, RNN and attention-based models, we use a $1$-$D$ convolution with a large stride as an additional preprocessing step to implement an operator akin to patching (Nie et al., 2023) and facilitate the processing at subsequent layers. The spatial module, when used, is implemented as a simple spatial attention layer (denoted as **sp. attn.**). For additional details, refer to App. C.

**Benchmarks** We use 4 real-world datasets from different domains, widely used in the time series forecasting literature (Zhang and Yan, 2023; Liu et al., 2023; Zeng et al., 2023; Nie et al., 2023; Wang et al., 2024): **Electricity** collects hourly electricity usage for 321 customers (Wu et al., 2021); **Weather** includes 21 meteorological variables collected every 10 minutes from Germany (Wu et al., 2021); **Traffic** contains hourly road occupancy data collected from 862 sensors in San Francisco (Wu et al., 2021); **Solar** contains 10-minute records of solar power generation from 137 photovoltaic plants (Lai et al., 2018). We split the data $70\%/10\%/20\%$ for the training, validation, and testing, following previous works (Wang et al., 2024). Metrics are computed on scaled data for consistency with published benchmarks. All the results report the standard deviations across 3 independent runs with different random seeds. We use an input window of 96 for all experiments in the main body of the paper, while in Tab. 3 we use a longer window size of 336 (except for Solar). For further details on the

---

[2]https://github.com/valentina-moretti/
deep-forecasting-benchmarking-issues

hyperparameters, refer to App. D.

# 3. What Matters in Deep Learning for Time Series Forecasting?

We examine four key design dimensions that characterize forecasting architectures and strongly influence overall performance. We focus on how these dimensions have been addressed in recent research and argue that the common practice of integrating design choices into architectures without explicitly evaluating their impact has contributed to frequent, unexpected empirical results. We support our position with plenty of empirical evidence in each section; extensive additional results are reported in Appendix E. We emphasize that the experiments are not intended to establish a ranking of existing models on benchmarks, but rather to provide evidence supporting our position. This section is structured around the following four design dimensions:

**D1. Model configuration** refers to the type of forecasting model. We distinguish among local, global, and hybrid approaches that combine elements of both paradigms.

**D2. Preprocessing and exogenous variables** covers data transformations and exogenous variables used as additional inputs to the forecasting architecture.

**D3. Temporal processing** includes the operators used to model temporal dependencies within the architecture.

**D4. Spatial processing** covers mechanisms modeling inter-series dependencies when multiple input time series are available.

## 3.1. Design Dimension 1: Model Configuration

As previously discussed, the model configuration–global, local, or hybrid– is a fundamental aspect in model design, since it radically changes the type of model being used. Yet, it is often left unspecified or dealt with as an implementation detail. However, choosing between a local, global, or hybrid approach has several implications that should be properly discussed (Salinas et al., 2020; Montero-Manso and Hyndman, 2021; Januschowski et al., 2020). For instance, as mentioned in Sec. 1, it has been common to model any collection of synchronous time series as a single highly-dimensional multivariate time series and hence consider models such as

$$\widehat{\boldsymbol{X}}_{t:t+H} = \mathcal{F}\left(\boldsymbol{X}_{t-W:t}, \ldots; \boldsymbol{\theta}\right). \quad (2)$$

However, this approach scales poorly with the input's dimensionality. Indeed, recent works (e.g., Nie et al. 2023; Liu et al. 2023) have observed that processing each channel independently with the same parameters empirically yields

*Table 1.* Comparison (MSE) of models with local embeddings for a forecasting horizon of 96. Best average results are in **bold**.

| D | Model | Hybrid | Global |
|---|---|---|---|
| Electr. | Transf. | $\mathbf{0.136}_{\pm\mathbf{.000}}$ | $0.151_{\pm.000}$ |
| | Crossformer | $\mathbf{0.141}_{\pm\mathbf{.001}}$ | $0.146_{\pm.003}$ |
| | TimeMixer | $\mathbf{0.151}_{\pm\mathbf{.000}}$ | $0.180_{\pm.001}$ |
| | iTransformer | $\mathbf{0.139}_{\pm\mathbf{.000}}$ | $0.154_{\pm.000}$ |
| Weather | Transf. | $\mathbf{0.153}_{\pm\mathbf{.001}}$ | $0.177_{\pm.002}$ |
| | Crossformer | $\mathbf{0.154}_{\pm\mathbf{.003}}$ | $0.164_{\pm.003}$ |
| | TimeMixer | $\mathbf{0.164}_{\pm\mathbf{.002}}$ | $0.178_{\pm.001}$ |
| | iTransformer | $\mathbf{0.154}_{\pm\mathbf{.000}}$ | $0.170_{\pm.001}$ |
| Traffic | Transf. | $0.417_{\pm.009}$ | $\mathbf{0.392}_{\pm\mathbf{.000}}$ |
| | Crossformer | $0.540_{\pm.014}$ | $\mathbf{0.512}_{\pm\mathbf{.007}}$ |
| | TimeMixer | $0.464_{\pm.001}$ | $\mathbf{0.463}_{\pm\mathbf{.001}}$ |
| | iTransformer | $0.435_{\pm.002}$ | $\mathbf{0.409}_{\pm\mathbf{.000}}$ |
| Solar | Transf. | $\mathbf{0.196}_{\pm\mathbf{.000}}$ | $0.205_{\pm.001}$ |
| | Crossformer | $0.177_{\pm.008}$ | $\mathbf{0.166}_{\pm\mathbf{.005}}$ |
| | TimeMixer | $\mathbf{0.366}_{\pm\mathbf{.017}}$ | $0.367_{\pm.017}$ |
| | iTransformer | $\mathbf{0.189}_{\pm\mathbf{.001}}$ | $0.197_{\pm.002}$ |

better performance. This corresponds to the well-known global approach, i.e., to processing related time series as

$$\widehat{\boldsymbol{x}}_{t:t+H}^i = \mathcal{F}\left(\boldsymbol{x}_{t-W:t}^i, \ldots; \boldsymbol{\theta}\right) \quad \forall i = 1, \ldots, N. \quad (3)$$

Moreover, several architectures, e.g., (Zhang and Yan, 2023; Wang et al., 2024; Donghao and Xue, 2024), adopt the approach in Eq. 3, but introduce some time series specific parameters $\phi^i$, resulting in hybrid global-local models (Smyl, 2020; Benidis et al., 2022; Cini et al., 2023):

$$\widehat{\boldsymbol{x}}_{t:t+H}^i = \mathcal{F}\left(\boldsymbol{x}_{t-W:t}^i, \ldots; \boldsymbol{\theta}, \boldsymbol{\phi}^i\right) \quad \forall i = 1, \ldots, N. \quad (4)$$

However, this choice is often not explicitly discussed in the respective papers and can only be determined by examining the code. For instance, we found that Wang et al. (2024) uses learnable local parameters in the normalization module; Salinas et al. (2020)–while relying on an otherwise global model–uses a different one-hot-encoding vector associated with each processed time series, effectively introducing a vector of learnable parameters specific to that input sequence. Other approaches–often relying on simple (linear) models (Zeng et al., 2023)–design models in Eq. 2 with separate parameters for each time series

$$\widehat{\boldsymbol{x}}_{t:t+H}^i = \mathcal{F}\left(\boldsymbol{x}_{t-W:t}^i, \ldots; \boldsymbol{\theta}^i\right) \quad \forall i = 1, \ldots, N, \quad (5)$$

hence yielding *local models*. Clearly, models in Eq. 2–5 represent fundamentally different approaches that can result in markedly different performance. Ignoring the impact of the associated design choices can be problematic for several reasons. First, the use of shared versus local parameters may have very different effects depending on whether the time series are homogeneous (e.g., data from identical

sensors at different locations) or heterogeneous (e.g., measurements of different physical quantities). Moreover, when dealing with multiple multivariate time series, a multivariate global model is often preferable to a univariate one that processes channels independently. Second, as we will see, **comparing the results of models belonging to different families without stating it explicitly can make it difficult to interpret performance differences**. For example, comparing an architecture designed for an inductive setting with one evaluated in a transductive setting might disadvantage the first, as the inductive models may be subject to additional constraints introduced to handle unseen time series. In Tab. 1, we assess the effect of local parameters on the performance–in terms of mean square error (MSE)–of different architectures on standard benchmarks (see Sec. 2.2) in long-range time series forecasting. We evaluate changes in performance for the reference Transformer and two architectures that incorporate local parameters by removing these components, and, conversely, for the iTransformer by adding them. As one would expect, using local parameters drastically changes results: mixing findings from the two columns of Tab. 1 without accounting for this–as is common in the literature–leads to misleading conclusions.

> **Key takeaway: when comparing forecasting architectures, the model configuration–global, local, or hybrid–must be carefully considered**; a given model configuration can be inherently advantaged over another due to experimental setting conditions (e.g., large vs small sample size, homogeneous vs heterogeneous time series, inductive vs transductive setting, etc.).

## 3.2. Design Dimension 2: Preprocessing and Exogenous Variables

Exogenous variables and preprocessing (e.g., scaling, detrending, and methods accounting for seasonality) are ingredients that can strongly affect performance. Popular architectures adopt many distinct choices regarding both preprocessing and the handling of exogenous variables. However, existing benchmarks often compare them directly without accounting for these differences. Similar to model configuration, these benchmarking practices further prevent a clear understanding of the reasons behind the observed performances. Moreover, recent papers often compare newly proposed architectures directly against published results of existing methods, without reproducing them. This practice makes it even more difficult to account for differences in preprocessing routines. To show the extent of this problem in recent benchmarks, we focus on exogenous variables. For instance, PatchTST, DLinear, and Crossformer do not use covariates in the original implementations, while iTransformer does. In Tab. 2, we show the impact of including the same covariates (calendar features, in this case) to DLin-

*Table 2.* Comparison (MSE) of models with and without covariates for a forecasting horizon of 96. Best average results are in **bold**

| D | Model | w/ exog. | w/out exog. |
|---|---|---|---|
| Electr. | Transf. | $0.136_{\pm.000}$ | $0.155_{\pm.001}$ |
| | PatchTST | $0.128_{\pm.000}$ | $0.134_{\pm.000}$ |
| | Crossformer | $0.139_{\pm.002}$ | $0.141_{\pm.001}$ |
| | iTransformer | $0.154_{\pm.000}$ | $0.167_{\pm.000}$ |
| | DLinear | $0.193_{\pm.000}$ | $0.195_{\pm.000}$ |
| Weather | Transf. | $0.153_{\pm.001}$ | $0.161_{\pm.000}$ |
| | PatchTST | $0.174_{\pm.000}$ | $0.180_{\pm.002}$ |
| | Crossformer | $0.154_{\pm.003}$ | $0.154_{\pm.003}$ |
| | iTransformer | $0.170_{\pm.001}$ | $0.176_{\pm.000}$ |
| | DLinear | $0.199_{\pm.005}$ | $0.196_{\pm.001}$ |
| Traffic | Transf. | $0.417_{\pm.009}$ | $0.479_{\pm.006}$ |
| | PatchTST | $0.355_{\pm.000}$ | $0.383_{\pm.001}$ |
| | Crossformer | $0.548_{\pm.024}$ | $0.540_{\pm.014}$ |
| | iTransformer | $0.409_{\pm.000}$ | $0.444_{\pm.001}$ |
| | DLinear | $0.609_{\pm.000}$ | $0.648_{\pm.000}$ |
| Solar | Transf. | $0.196_{\pm.000}$ | $0.206_{\pm.003}$ |
| | PatchTST | $0.196_{\pm.001}$ | $0.225_{\pm.003}$ |
| | Crossformer | $0.176_{\pm.006}$ | $0.177_{\pm.008}$ |
| | iTransformer | $0.197_{\pm.002}$ | $0.221_{\pm.003}$ |
| | DLinear | $0.246_{\pm.001}$ | $0.285_{\pm.001}$ |

ear, PatchTST, and Crossformer, and removing them from iTransformer and the reference Transformer. Depending on the dataset, their effects can drastically affect performance, yet benchmarks often compare models that include exogenous variables to those that do not.

> **Key takeaway: all methods being compared must be provided with the same input data, including any exogenous variables that can inform the downstream task**. Results from new experiments must not be combined with those existing in the literature unless it is possible to ensure an identical evaluation setup (e.g., through a third-party benchmarking platform).

## 3.3. Design Dimension 3: Temporal Processing

This design dimension, concerning sequence modeling operators, has been the main focus of recent research. However, this line of work has produced contrasting results, leading to considerable confusion about which components truly drive performance (Zeng et al., 2023; Toner and Darlow, 2024; Tan et al., 2024). Current trends in the field focus on finding a one-size-fits-all architecture with SOTA performance in benchmarks. This prompted the adoption of increasingly complex architectures stacking multiple components whose effectiveness often relies on hidden implementation details. Since current benchmarks do not account for these differences (as shown in Sec. 3.1 and Sec. 3.2), tracking the source of performance gains becomes increasingly diffi-

*Table 3.* Forecasting results (MSE and MAE) for a horizon of 96 steps for models *not including* spatial processing. Best average results are in **bold**, second best are underlined.

| MODEL | Electricity | | Weather | | Traffic | | Solar | |
|---|---|---|---|---|---|---|---|---|
| | MSE | MAE | MSE | MAE | MSE | MAE | MSE | MAE |
| Linear Global | 0.140 | 0.237 | 0.174 | 0.234 | 0.410 | 0.282 | 0.222 | 0.291 |
| Linear Local | 0.134 | 0.230 | **0.144** | 0.209 | 0.426 | 0.298 | 0.223 | 0.295 |
| MLP | $0.129_{\pm.000}$ | $0.225_{\pm.000}$ | $0.148_{\pm.001}$ | $0.198_{\pm.000}$ | $0.376_{\pm.000}$ | $0.253_{\pm.001}$ | $0.194_{\pm.003}$ | $0.239_{\pm.002}$ |
| RNN | $0.147_{\pm.001}$ | $0.247_{\pm.001}$ | $0.149_{\pm.001}$ | $0.203_{\pm.001}$ | $0.390_{\pm.007}$ | $0.275_{\pm.002}$ | $0.200_{\pm.003}$ | $0.246_{\pm.004}$ |
| TCN | $0.130_{\pm.000}$ | $0.224_{\pm.000}$ | $0.148_{\pm.000}$ | $0.200_{\pm.001}$ | $0.364_{\pm.003}$ | $0.253_{\pm.002}$ | $\underline{0.193}_{\pm.004}$ | $0.243_{\pm.005}$ |
| Transf. | $0.129_{\pm.001}$ | $0.222_{\pm.001}$ | $0.149_{\pm.001}$ | $0.203_{\pm.002}$ | $\underline{0.362}_{\pm.003}$ | $\underline{0.249}_{\pm.002}$ | $0.203_{\pm.006}$ | $0.245_{\pm.002}$ |
| Pyraf. | $\underline{0.129}_{\pm.001}$ | $0.224_{\pm.001}$ | $0.148_{\pm.001}$ | $0.199_{\pm.001}$ | $0.365_{\pm.002}$ | $0.251_{\pm.003}$ | $\mathbf{0.189}_{\pm.003}$ | $\mathbf{0.236}_{\pm.004}$ |
| TimeMixer | $0.129_{\pm.001}$ | $0.224_{\pm.000}$ | $\underline{0.147}_{\pm.001}$ | $\underline{0.197}_{\pm.000}$ | $0.373_{\pm.002}$ | $0.271_{\pm.003}$ | $0.199_{\pm.001}$ | $0.245_{\pm.000}$ |
| PatchTST | $\mathbf{0.125}_{\pm.000}$ | $\mathbf{0.218}_{\pm.000}$ | $0.148_{\pm.001}$ | $\mathbf{0.195}_{\pm.001}$ | $\mathbf{0.345}_{\pm.000}$ | $\mathbf{0.234}_{\pm.000}$ | $0.197_{\pm.001}$ | $0.244_{\pm.004}$ |
| DLinear | $0.140_{\pm.000}$ | $0.237_{\pm.000}$ | $0.173_{\pm.000}$ | $0.232_{\pm.001}$ | $0.407_{\pm.000}$ | $0.283_{\pm.000}$ | $0.246_{\pm.001}$ | $0.331_{\pm.000}$ |

*Table 4.* Forecasting results for a horizon of 96 steps for models *including* spatial processing. Best average results are in **bold**, second best are underlined.

*(a)* Comparison (MSE and MAE) of models *including* spatial processing.

| Model | Electricity | | Weather | | Traffic | | Solar | |
|---|---|---|---|---|---|---|---|---|
| | MSE | MAE | MSE | MAE | MSE | MAE | MSE | MAE |
| MLP + sp. attn. | $0.140_{\pm.001}$ | $0.238_{\pm.001}$ | $0.157_{\pm.000}$ | $\underline{0.202}_{\pm.001}$ | $0.435_{\pm.006}$ | $0.275_{\pm.001}$ | $0.201_{\pm.009}$ | $0.246_{\pm.003}$ |
| Pyraf. + sp. attn. | $0.139_{\pm.001}$ | $0.236_{\pm.001}$ | $0.157_{\pm.002}$ | $0.204_{\pm.001}$ | $\mathbf{0.389}_{\pm.002}$ | $\underline{0.267}_{\pm.001}$ | $0.188_{\pm.002}$ | $0.235_{\pm.003}$ |
| iTransformer | $0.148_{\pm.000}$ | $0.241_{\pm.000}$ | $0.171_{\pm.001}$ | $0.210_{\pm.001}$ | $\underline{0.393}_{\pm.001}$ | $\mathbf{0.266}_{\pm.001}$ | $0.208_{\pm.003}$ | $0.240_{\pm.006}$ |
| Crossformer | $\mathbf{0.136}_{\pm.000}$ | $\mathbf{0.232}_{\pm.001}$ | $\mathbf{0.152}_{\pm.003}$ | $0.222_{\pm.004}$ | $0.527_{\pm.002}$ | $0.270_{\pm.003}$ | $\mathbf{0.184}_{\pm.008}$ | $\underline{0.227}_{\pm.006}$ |
| ModernTCN | $0.141_{\pm.000}$ | $0.237_{\pm.001}$ | $\underline{0.154}_{\pm.001}$ | $\mathbf{0.200}_{\pm.001}$ | $0.445_{\pm.001}$ | $0.287_{\pm.001}$ | $0.190_{\pm.001}$ | $\mathbf{0.222}_{\pm.002}$ |

*(b)* Ablation (MSE): iTransformer with or without space attention.

| | iTransformer | |
|---|---|---|
| Dataset | Space att. | Feedforward |
| Electricity | $\mathbf{0.148}_{\pm.000}$ | $0.149_{\pm.001}$ |
| Weather | $\mathbf{0.171}_{\pm.001}$ | $\mathbf{0.171}_{\pm.000}$ |
| Traffic | $0.393_{\pm.001}$ | $\mathbf{0.390}_{\pm.001}$ |
| Solar | $0.208_{\pm.003}$ | $\mathbf{0.194}_{\pm.001}$ |

cult as model architectures grow more complex. We focus on methods that process inputs only along the temporal dimension, while approaches that model spatial dependencies are discussed in Sec. 3.4. In Tab. 3, we compare the reference architectures introduced in Sec. 2.2 against three popular and well-established baselines–DLinear, PatchTST, and TimeMixer–using standardized inputs (including covariates) and hyperparameter tuning. **Results show that no single model consistently outperforms the others and that reference architectures relying on standard and simple operators achieve competitive performance against the SOTA across all the considered scenarios**. Note that we do not aim to identify the best architecture, but to show that other design choices–beyond the sequence modeling operators–can strongly influence observed performance. The fact that simple models, properly configured as hybrid global-local with exogenous inputs, can match the SOTA shows that benchmarks often credit sequence modeling operators for performance gains that actually stem from other design choices. Analogous observations are confirmed in Sec. 3.4. Additionally, these results further support our observations of Sec. 3.1. In particular, all the models in our experiments, apart from the Linear Local (a local OLS linear model), treat the Weather dataset as a collection of univariate time series, reflecting how this dataset is commonly handled in the literature. Interestingly, one of the best-performing models on

Weather is actually the local OLS Linear model. This aligns with our analysis, since Weather actually is a multivariate time series with heterogeneous channels, and among the models in Tab. 3, the local OLS Linear model is the only one that explicitly models each time series as heterogeneous. Finally, Fig. 1a reports additional aspects that should be considered in comparisons, namely the computational scalability of the architectures–in terms of batch processing time and GPU memory usage–in relation to forecasting accuracy.

> **Key takeaway: evaluating the effect of a sequence modeling operator on performance requires keeping all other design dimensions identical among the models being compared**; otherwise, any potential gain deriving from variations in other design dimensions may be incorrectly attributed to the operator.

### 3.4. Design Dimension 4: Spatial Processing

We call *spatial* the dimension that spans multiple time series, which may correspond to different spatial locations when considering physical sensors. We complement the discussion started in Sec. 3.3 by considering models that account for inter-series dependencies using different operators. We compare the reference architectures, where dependencies are modeled with a standard spatial Transformer,

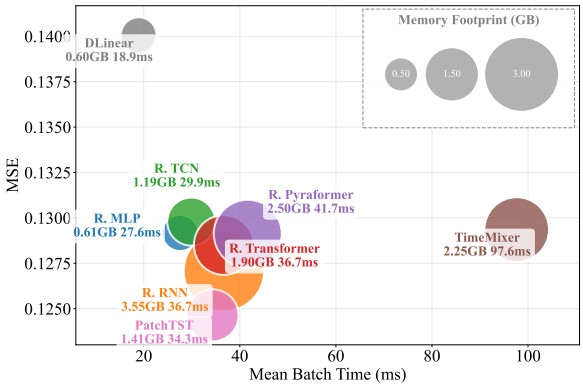

*(a)* Models *not including* spatial processing.

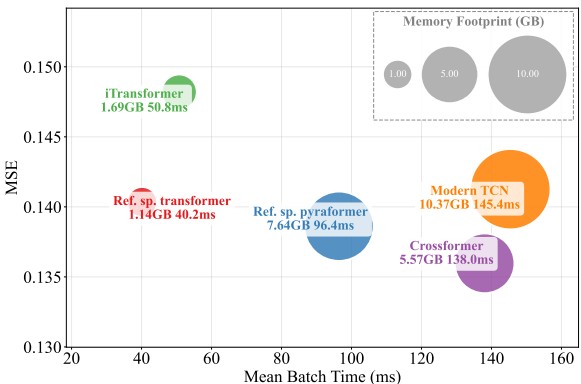

*(b)* Models *including* spatial processing.

*Figure 1.* MSE versus mean batch time during training on the Electricity dataset for a forecasting horizon of 96. Circle size indicates memory consumption.

against three SOTA baselines: iTransformer, Crossformer, and ModernTCN. The reference architectures use an MLP or pyramidal attention for temporal processing. Tab. 4a reports the results of the comparison, where we reduced the length of the input window to keep computational costs manageable. As in Sec. 3.3, simulations show that simple, streamlined architectures perform comparably to the SOTA, highlighting once again the limitations of current benchmarking practices. Moreover, considering results in Tab. 3 and Tab. 4a, we decided to test the effectiveness of spatial attention operators in this context further. For this reason, we introduce an additional baseline obtained by replacing the spatial attention layer in iTransformer with a simple MLP, removing all components modeling spatial dependencies in the architecture. Surprisingly, the results in Tab. 4b show that, in this context, **entirely removing spatial attention led to better or similar performance in all the considered datasets**. This result shows again the shortcomings of current benchmarking in attributing performance gains to specific components in SOTA architectures. Finally, Fig. 1b reports performance in relation to computational cost; this is critical since processing along the spatial dimension can have a great impact on scalability and, as such, it is impor-

tant to ensure that the increase in computational cost pays out in terms of forecasting accuracy.

> **Key takeaway: when a spatial component is introduced, it is necessary to evaluate its individual impact on performance**, e.g., through ablation studies; failing to do so can mislead the design of architectures toward elements that do not truly improve performance.

## 4. Call to Action

Results in Sec. 3 show that current benchmarks fail to factor out specific design differences among compared architectures and may incorrectly attribute performance gains that instead are due to other aspects of the design space. This can undermine our understanding and steer research in the wrong direction, ultimately hindering progress in the field. To address the issues, we believe that two steps are particularly necessary.

**First, we call for benchmarks that account for key design dimensions and isolate the effects of the proposed designs on performance**. When comparing architectures that differ along one key design dimension (e.g., a global model against a hybrid approach), such differences should be openly and explicitly discussed as part of the analysis. When proposing a new design w.r.t. a specific dimension, it should be evaluated compartmentally, by keeping all other design choices fixed, e.g., as in Tab. 3. For instance, the performance of a newly proposed temporal processing block must be assessed against baselines while keeping the same model configuration, input data, and spatial processing (if any). Failing to do so would inevitably introduce the risk of misassigning credit to newly introduced designs. Instead, we advocate for explicitly accounting for differences across other design dimensions in the compared architectures when attributing performance gains to specific components, regardless of overall model complexity. This requires environments that allow effectively to test the single contributions. In this direction, a promising path is the development of benchmarks that isolate key design dimensions, possibly by relying on synthetic datasets, which help measure the individual effects of different components. Moreover, understanding the actual contribution of different design dimensions should also involve appropriate statistical comparisons to ensure that observed performance gains are significant.

**Second, we call to explicitly report relevant design choices in model documentation**. As noted in Sec. 3.3, many critical design choices are often left unspecified in published work or treated as implementation details. Our results show that these "hidden" choices can significantly affect performance and produce misleading outcomes. Documenting key forecasting design choices clarifies model architectures and ensures accurate interpretation of results.

## FORECASTING MODEL CARD

**Model setting**
- Size of the input window
- Whether the model is transductive or inductive, and can be used in a cold start scenario
- How to mask missing observations and/or if imputation is needed

**D1. Model configuration**
- Whether the model is global, local, or hybrid
- *If the model is hybrid*, which parameters are shared across the time series and which are not

**D2. Preprocessing and exogenous variables**
- The type of scaling or other transformation applied at training and inference time
- Temporal covariates, lagged variables, or other types of exogenous variables employed

**D3. Temporal processing**
- Modules and operators used to encode observations along the temporal axis
- Time and space complexity w.r.t. the length of the time series being processed

**D4. Spatial processing** *If spatial dependencies are accounted for:*
- Modules used to model spatial dynamics and whether a graph structure is employed
- Time and space complexity w.r.t. the number of the time series being processed

*Figure 2.* Forecasting Model Card

Inspired by model cards (Mitchell et al., 2019), we propose an *auxiliary forecasting model card* (Fig. 2) tailored to the design dimensions discussed in this work. An example of its use is shown in App. G. The forecasting model card provides a structured framework for documentation, promoting transparency, reproducibility, and more informed comparisons across forecasting methods, and helping users to understand the model characteristics. Moreover, it not only reduces the time and effort to extract such information from code or supplementary materials but also limits errors and ambiguities that may result from this exercise. It can also guide the design of ablation studies. The lack of a good ablation study is indeed often a consequence of overlooking design aspects that model cards make explicit.

Revisiting evaluation practices to reliably measure progress would refocus research on foundational methodological questions, ultimately yielding more *robust* and *interpretable* advances in time series forecasting.

## 5. Alternative Views

Alternative views to our position are presented below.

**The problem is only in the quality of the data and the fairness of the evaluation procedure.** Common benchmarks have been criticized for relying on a narrow set of often saturated datasets, where models achieve marginal, potentially insignificant performance gains, often stemming from overfitting rather than methodological improvements (Wang et al., 2025; Shchur et al., 2025). Several recent works have then focused on introducing more challenging datasets and exhaustive benchmarks (Han et al., 2026; Tan et al., 2025). Moreover, prior studies have also shown that current practices often involve inconsistent hyperparameter tuning procedures and biased selection of the evaluation setup and baselines (Eftimov et al., 2022; Roque et al., 2025; Brigato et al., 2026). Therefore, one might argue that the observed contradictory results (Toner and Darlow, 2024; Zeng et al., 2023) stem from these issues alone and that fixing them would be enough to ensure progress. We do acknowledge that these aspects play a central role and that correctness, in particular, is the first necessary step toward any meaningful discussion. Nonetheless, we argue that data quality and consistent evaluation alone are not enough. Even with better datasets and consistent model selection, failing to factor out the impact of different designs can distort performance assessments and lead to wrong conclusions. Indeed, our experiments show how issues in our benchmarking practices go beyond fair hyperparameter tuning and data quality. These considerations are also reflected in the next alternative view, which, we argue, suffers from similar limitations.

**Under a fair and standardized evaluation, benchmarking complex architectures, even without analyzing specific components, can drive progress efficiently.** Benchmarks have been very effective in stimulating progress in machine learning research–think of Imagenet (Deng et al., 2009) and the CASP experiment (Kryshtafovych et al., 2021; Jumper et al., 2021). In time series forecasting, the M competitions (Makridakis et al., 2020; 2022) have played a similar role, for instance, showcasing the effectiveness of the global approach (Smyl, 2020). One could argue, then, that chasing performance on established benchmarks–which evaluate architectures as a whole–is an efficient and effective way to ensure progress. From this perspective, comparing forecasting architectures on representative datasets under fair and standardized evaluation procedures would be enough to support continued advancement. Although this position has merits, e.g., in cases where isolating the contribution of individual components is not straightforward, we have shown that the resulting approach is extremely brittle. Indeed, we believe that focusing on the final performance of monolithic architecture inherently limits our understanding and can lead to pitfalls. Conversely, analyzing how

different components behave in different scenarios, and to what degree different designs affect performance in such settings, leads to actionable and possibly transferable insights. This can also be pursued when design dimensions cannot be fully disentangled, e.g., by explicitly acknowledging architectural differences and assessing their impact through appropriate baselines and ablation studies. Even if their individual effects cannot be completely isolated, making these differences explicit helps avoid misleading conclusions. We believe this approach is key to providing practitioners with guidelines and increasing trust in the results produced by the research community.

## 6. Conclusion

In this paper, we argue that current benchmarking practices in time series forecasting fail to accurately measure progress in the field. We show that benchmarks overlook crucial design differences among compared models, producing misleading results. Our position is that meaningful advancement in time series forecasting requires benchmarks to explicitly account for key design dimensions. With this work, we aim to foster a discussion that will enable the field to move forward and address its current limitations.

**Limitations**   In this work, we focus on a specific set of dimensions within the design space of forecasting models that strongly influence empirical results of recent studies. The analysis can be extended to additional baselines and forecasting settings, such as probabilistic or short-term, and to alternative metrics. Nevertheless, our evaluation already provides clear evidence highlighting the shortcomings of current benchmarking practices in advancing the field. Finally, the emerging area of foundational models for time series has recently demonstrated striking results: the benchmarking issues discussed throughout this work are also directly relevant in this context. Extending the present analysis to foundation models by possibly identifying additional, relevant design dimensions represents an important direction for future discussion.

## Acknowledgments

This work was supported by the Swiss National Science Foundation projects no. 204061 (High-Order Relations and Dynamics in Graph Neural Networks) and no. 225351 (Relational Deep Learning for Reliable Time Series Forecasting at Scale).

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

# Appendix

## A. Extended Context

The long history of neural networks in forecasting applications has often been characterized by skepticism (Zhang et al., 1998). The forecasting community has reached consensus on the effectiveness of deep learning methods when a single neural network can be trained on (large) collections of related time series (Hewamalage et al., 2021; Benidis et al., 2022). Models following this approach are called *global* in contrast with *local* models trained separately on each time series (Montero-Manso and Hyndman, 2021; Januschowski et al., 2020; Benidis et al., 2022). Global models and hybrid global-local variants thereof have won forecasting competitions (Smyl, 2020) and been adopted by the industry (Salinas et al., 2020; Kunz et al., 2023). As new sequence modeling architectures gain popularity (Vaswani et al., 2017; Gu et al., 2022; Orvieto et al., 2023; Gu and Dao, 2024), the machine learning community has started investigating how to adapt them to forecasting. In particular, the Informer (Zhou et al., 2021) was among the first architectures tailoring Transformers (Vaswani et al., 2017) to long-range time series forecasting. Together with the architecture, Zhou et al. (2021) also introduced a popular benchmark where collections of time series are modeled as a single multivariate sequence. However, conflating the problem of forecasting any group of time series into forecasting a single multivariate sequence can be problematic and lead to unclear designs (Sec. 3.1). Several subsequent works followed the same setup (Wu et al., 2021; Liu et al., 2022a;b; Zhou et al., 2022; Wu et al., 2023). Then, Zeng et al. (2023) and Toner and Darlow (2024) showed that simple linear models outperform most of these architectures in such benchmarks. At the same time, Nie et al. (2023) showed – again in the same setup – that superior results could be achieved by processing each channel independently with shared parameters. However, for many of these benchmarks, this essentially corresponds to the already well-understood global approach, as most of the associated datasets consist of collections of related time series. Follow-up works (Liu et al., 2023; Zhang and Yan, 2023) reintroduced *spatial* components to model dependencies across multiple variates while keeping the core of the model global; this had already been studied in depth in the context of spatiotemporal forecasting, e.g., with graph-based architectures (Jin et al., 2024; Cini et al., 2025). The aforementioned models, beyond the proposed method, rely on several different design choices. Therefore, as shown in Sec. 3, directly comparing them, as commonly done in benchmarks, can lead to unexpected and contradictory results (Toner and Darlow, 2024; Zeng et al., 2023; Tan et al., 2024).

## B. Baselines

Below, we provide a brief description of each baseline as employed in our experiments on the considered benchmarks. Furthermore, we summarize them in Tab. 5 using three fields corresponding to the design dimensions introduced in Sec. 3, excluding the *preprocessing and exogenous variables* dimension due to the considerable differences among the methods.

- Dlinear (Zeng et al., 2023) decomposes the input into seasonal and trend components using a moving average and processes them with linear layers. The hyperparameters determine its local-global nature. In the table, we report it as global because, in our experiments, it was used in this configuration. We follow the same convention for PatchTST and TimeMixer.

- PatchTST (Nie et al., 2023) has strongly influenced subsequent works by employing a global Transformer, in contrast to earlier local multivariate approaches that treated the group of input time series as a single multivariate series. PatchTST segments the time series and generates corresponding embeddings using an operation analogous to temporal convolution. Then, it applies attention over these segments, referred to as *patches*. It does not model spatial relations.

- TimeMixer (Wang et al., 2024) is a fully MLP-based architecture that downsamples the input at different scales, decomposes it into trend and seasonal components, and employs feedforward layers to model temporal dependencies.

- Crossformer (Zhang and Yan, 2023) employs an input encoding with segmentation analogous to that used in PatchTST. The model is a hybrid global-local model, as it includes learnable position embeddings for each time series in the set. In addition to temporal attention, it captures spatial dependencies through attention over the spatial dimension using a routing mechanism. Furthermore, it adopts a hierarchical encoder-decoder structure.

- iTransformer (Liu et al., 2023) is a global model that uses a feedforward approach to encode temporal dynamics and spatial attention to model spatial dependencies. This method has been described as applying attention to the *inverted dimension*, i.e., the spatial dimension. The model is global.

*Table 5.* Description for the baseline models

| Model | Model config. | Temporal processing | Spatial processing |
|---|---|---|---|
| Dlinear | Global | Linear layers | Not modeled |
| PatchTST | Global | Temporal convolution followed by temporal attention over the patches | Not modeled |
| TimeMixer | Hybrid | Feedforward networks applied to the trend and seasonal components, downsampled at different scales | Not modeled |
| Crossformer | Hybrid | Temporal convolution followed by attention applied over the patches, with a hierarchical structure constructed with linear layers | Spatial attention applied among patches of different time series |
| iTransformer | Global | Feedforward layers | Spatial attention applied among different time series |
| ModernTCN | Hybrid | Depth-wise convolutions | Convolution applied across time series |
| Linear global/local | Global/local | Linear autoregression | Not modeled |

- ModernTCN (Donghao and Xue, 2024) uses depth-wise convolutions to encode temporal information, with an encoding similar to that performed in PatchTST, and then applies point-wise convolutions to process the feature and spatial dimensions separately.

- Linear (Toner and Darlow, 2024) is linear autoregressive models trained with $L2$ regularization and OLS. The *local* variant employs different weights for each series, while the *global* variant employs the same weights for all the series in the set.

## C. Reference Architectures Structure

The reference architectures, as schematized in Fig. 3, consist of a preprocessing module, followed by the processing of the temporal and spatial dynamics, and finally a postprocessing module. Their modular structure facilitates understanding of the architecture and promotes fair comparisons, as each module can be modified independently. In our experiments, we kept most modules fixed, modifying only the temporal and spatial modules for the experiments reported in Sec. 3.3 and 3.4, respectively, and occasionally the feature encoding module. Here, we provide a more detailed description, complementing Sec. 2.2.

**Preprocessing module** The preprocessing module begins with RevInv (Kim et al., 2022) normalization. Then, the feature encoding module processes the input and covariates through non-linear layers and returns their sum. Alternatively, it can perform temporal convolution to generate an encoding similar to that in (Nie et al., 2023). Finally, local embeddings are concatenated with the resulting encoding.

**Processing module** The processing module consists of temporal processing followed by spatial processing. In a more general architecture, these components could be interleaved. For simplicity, however, they are treated separately in our implementation.

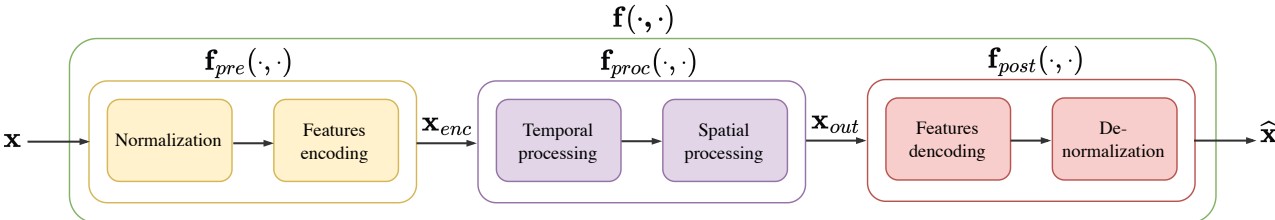

*Figure 3.* Block diagram of the reference architectures

*Table 6.* Information on the datasets.

| Dataset | Time series | Steps | Frequency | Domain |
|---|---|---|---|---|
| Weather | 21 | 52695 | 10min | Weather |
| Solar-Energy | 137 | 52559 | 10min | Energy |
| ECL | 321 | 26303 | Hourly | Electricity |
| Traffic | 862 | 17543 | Hourly | Transportation |

**Postprocessing module**   The postprocessing module consists of a linear decoder that maps the hidden representations to predictions for the horizon. Finally, the predictions are de-normalized using the RevInv module.

## D. Hyperparameter Tuning

For each experiment, we set a fixed batch size for each dataset. The hidden size is tuned between 32 and 256 for all datasets, with the addition of 16 for the Weather dataset. For the Tables and Figures in Sec. 3.3, Sec. 3.4, Tab. 10 –Tab. 16 and Fig. 4, Fig. 5 we used the hyperparameters of the best-performing configuration identified during tuning for each considered window size and for a 96-step forecasting horizon. Instead, hyperparameters in Tab. 1, Tab. 2, Tab. 7 and Tab. 9 are fixed and identical for both sides of the comparison. In Tab. 3, Tab. 11, and Fig. 4, the window size was set to 336 for all datasets except Solar. Throughout the paper, unless otherwise specified, the window size and the forecasting horizon are set to 96.

## E. Empirical Setup and Additional Experiments

In this section, we provide further details on the experiments conducted in Sec. 3.1–Sec. 3.4. Moreover, we present additional experiments and extend the tables from the paper by including the mean absolute error (MAE) metric, additional forecasting horizons and window sizes. Dataset details are provided in Tab. 6.

### E.1. Empirical Setup and Additional Experiments for D1: Model Configuration

To obtain the global version of models that include local parameters in the experiments of Sec. 3.1, we remove such local components. For example, the global TimeMixer model is obtained by removing the learnable parameters from the normalization module, while the global versions of Crossformer and the reference Transformer are obtained by removing their local embeddings. Conversely, to create a hybrid version of an otherwise global model, we add per-series local embeddings, which are used as an additional input, as done for iTransformer. This approach is straightforward and can be applied to different architectures, as also shown by recent works (Cini et al., 2023; Shao et al., 2022). The results in Tab. 7 extend Tab. 1 to include both MSE and MAE. We added in Tab. 8, a comparison of all the possible configurations—local, global, and hybrid—for linear models. The simple hybrid variants are obtained by concatenating learnable local parameters to the input for DLinear, and by concatenating a one-hot encoding representing the time series for Linear. The results are consistent with the findings discussed in Sec. 3.1.

### E.2. Empirical Setup and Additional Experiments for D2: Preprocessing and Exogenous Variables

For the experiment in Sec. 3.2, covariates were removed from the models that originally used them in their implementations, such as the reference Transformer and iTransformer, and added to the models that did not include them, such as PatchTST, DLinear, and Crossformer. For all models, we used datetime features encoded as sinusoids as covariates. These were simply concatenated to the input. In particular, for ModernTCN, we directly concatenate the covariates to the input along the channel dimension. For DLinear, we concatenated the exogenous variables corresponding to the last time step within each context window to the input, since DLinear is not a sequence model. For PatchTST and Crossformer, we unfold the covariate tensor in the same way as the input to form patches, and then concatenate the last timestep of each covariate patch to the corresponding input patch. iTransformer, instead, already include covariates in their implementations, and so we use them as in the original code (the same applies to Timemixer in the experiments in E.3). Note that, for each architecture, we verified that the specific method used to concatenate covariates produced reasonable results compared to other reasonable alternatives (e.g., summing the covariates as positional encodings). The results in Tab. 9 extend Tab. 2 to include both MSE and MAE.

*Table 7.* Comparison (MSE and MAE) of models with and without local parameters for a horizon of 96 steps. Best average results are in **bold**.

| D | Model | hybrid | | global | |
|---|---|---|---|---|---|
| | | MSE | MAE | MSE | MAE |
| Electr. | Transf. | $\mathbf{0.136}_{\pm.000}$ | $\mathbf{0.231}_{\pm.000}$ | $0.151_{\pm.000}$ | $0.242_{\pm.000}$ |
| | Crossformer | $\mathbf{0.141}_{\pm.001}$ | $\mathbf{0.235}_{\pm.001}$ | $0.146_{\pm.003}$ | $0.240_{\pm.003}$ |
| | TimeMixer | $\mathbf{0.151}_{\pm.000}$ | $\mathbf{0.248}_{\pm.001}$ | $0.180_{\pm.001}$ | $0.268_{\pm.001}$ |
| | iTransformer | $\mathbf{0.139}_{\pm.000}$ | $\mathbf{0.236}_{\pm.001}$ | $0.154_{\pm.000}$ | $0.245_{\pm.000}$ |
| Weather | Transf. | $\mathbf{0.153}_{\pm.001}$ | $\mathbf{0.198}_{\pm.000}$ | $0.177_{\pm.002}$ | $0.215_{\pm.001}$ |
| | Crossformer | $\mathbf{0.154}_{\pm.003}$ | $0.225_{\pm.004}$ | $0.164_{\pm.003}$ | $\mathbf{0.224}_{\pm.007}$ |
| | TimeMixer | $\mathbf{0.164}_{\pm.002}$ | $\mathbf{0.208}_{\pm.001}$ | $0.178_{\pm.001}$ | $0.216_{\pm.001}$ |
| | iTransformer | $\mathbf{0.154}_{\pm.000}$ | $\mathbf{0.199}_{\pm.001}$ | $0.170_{\pm.001}$ | $0.211_{\pm.001}$ |
| Traffic | Transf. | $0.417_{\pm.009}$ | $0.278_{\pm.005}$ | $\mathbf{0.392}_{\pm.000}$ | $\mathbf{0.260}_{\pm.001}$ |
| | Crossformer | $0.540_{\pm.014}$ | $0.279_{\pm.007}$ | $\mathbf{0.512}_{\pm.007}$ | $\mathbf{0.259}_{\pm.004}$ |
| | TimeMixer | $0.464_{\pm.001}$ | $0.328_{\pm.003}$ | $\mathbf{0.463}_{\pm.001}$ | $\mathbf{0.327}_{\pm.003}$ |
| | iTransformer | $0.435_{\pm.002}$ | $\mathbf{0.275}_{\pm.000}$ | $\mathbf{0.409}_{\pm.000}$ | $0.277_{\pm.001}$ |
| Solar | Transf. | $\mathbf{0.196}_{\pm.000}$ | $\mathbf{0.243}_{\pm.001}$ | $0.205_{\pm.001}$ | $0.247_{\pm.002}$ |
| | Crossformer | $0.177_{\pm.008}$ | $0.215_{\pm.004}$ | $\mathbf{0.166}_{\pm.005}$ | $\mathbf{0.204}_{\pm.006}$ |
| | TimeMixer | $\mathbf{0.366}_{\pm.017}$ | $\mathbf{0.396}_{\pm.013}$ | $0.367_{\pm.017}$ | $0.396_{\pm.013}$ |
| | iTransformer | $\mathbf{0.189}_{\pm.001}$ | $\mathbf{0.240}_{\pm.004}$ | $0.197_{\pm.002}$ | $0.243_{\pm.001}$ |

## E.3. Empirical Setup and Additional Experiments for D3: Temporal Processing

Results in Sec. 3.3 were obtained through extensive hyperparameter tuning for each model, configured with exogenous inputs (as detailed in E.2) as hybrid global-local embeddings. The results in Tab. 10 extend Tab. 3 to a broader set of horizons (96, 192, 336, 720). We observe that increasing the forecasting horizon does not change the conclusions drawn in the corresponding sections, and results still show that, even for larger forecasting horizons, standard, streamlined architectures achieve performance comparable to current SOTA models. Tab. 11 shows the computational efficiency of the models for increasing horizons on the Electricity dataset. We employed the PyTorch Profiler (Paszke et al., 2019) to monitor GPU performance during training, specifically collecting the total CUDA execution time. Additionally, GPU memory usage was obtained using the PyG (Fey et al., 2025) function *get_gpu_memory_from_nvidia_smi*. To ensure a consistent evaluation, all measurements related to model performance (Tab. 11 and 17) were conducted on the same machine running Oracle Linux Server 8.8, equipped with an Intel Xeon E5-2650 v3 CPU @ 2.30 GHz 20 (2 x 10) cores, 128 GB of system RAM, and an NVIDIA A100-PCIe GPU with 40 GB of HBM2 memory. Finally, we summarize these results in Fig. 4, which illustrates the trade-off between model performance and computational efficiency in terms of training batch time and GPU memory

*Table 8.* Comparison (MSE and MAE) of Linear models in their local, global, and hybrid variants for a horizon of 96 steps. A dash (–) marks experiments beyond our computational budget. Best mean results are in **bold**.

| Dataset | Model | Hybrid | | Global | | Local | |
|---|---|---|---|---|---|---|---|
| | | MSE | MAE | MSE | MAE | MSE | MAE |
| Electr. | DLinear | $0.195_{\pm.000}$ | $0.277_{\pm.000}$ | $0.195_{\pm.000}$ | $0.277_{\pm.000}$ | $\mathbf{0.184}_{\pm.000}$ | $\mathbf{0.270}_{\pm.000}$ |
| | OLS | $0.194_{\pm.000}$ | $0.277_{\pm.000}$ | $0.195_{\pm.000}$ | $0.277_{\pm.000}$ | $\mathbf{0.184}_{\pm.000}$ | $\mathbf{0.270}_{\pm.000}$ |
| Weather | DLinear | $0.198_{\pm.001}$ | $0.254_{\pm.002}$ | $0.196_{\pm.001}$ | $0.248_{\pm.002}$ | $\mathbf{0.161}_{\pm.001}$ | $\mathbf{0.233}_{\pm.001}$ |
| | OLS | $0.196_{\pm.000}$ | $0.254_{\pm.000}$ | $0.195_{\pm.000}$ | $0.253_{\pm.000}$ | $\mathbf{0.161}_{\pm.000}$ | $\mathbf{0.233}_{\pm.000}$ |
| Traffic | DLinear | $0.648_{\pm.000}$ | $\mathbf{0.395}_{\pm.000}$ | $0.648_{\pm.000}$ | $\mathbf{0.395}_{\pm.000}$ | $0.647_{\pm.000}$ | $0.403_{\pm.000}$ |
| | OLS | - | - | $0.649_{\pm.000}$ | $\mathbf{0.396}_{\pm.000}$ | $0.647_{\pm.000}$ | $0.403_{\pm.000}$ |
| Solar | DLinear | $0.286_{\pm.000}$ | $0.375_{\pm.001}$ | $\mathbf{0.285}_{\pm.001}$ | $\mathbf{0.372}_{\pm.001}$ | $0.286_{\pm.000}$ | $0.376_{\pm.001}$ |
| | OLS | $\mathbf{0.285}_{\pm.000}$ | $\mathbf{0.372}_{\pm.000}$ | $\mathbf{0.285}_{\pm.000}$ | $\mathbf{0.372}_{\pm.000}$ | $0.286_{\pm.000}$ | $0.374_{\pm.000}$ |

*Table 9.* Comparison (MSE and MAE) of models with and without covariates for a horizon of 96 steps. Best average results are in **bold**.

| D | Model | w/ exog. | | w/o exog. | |
|---|---|---|---|---|---|
| | | MSE | MAE | MSE | MAE |
| Electr. | Transf. | **0.136**±.000 | **0.231**±.000 | 0.155±.001 | 0.247±.000 |
| | PatchTST | **0.128**±.000 | **0.222**±.000 | 0.134±.000 | 0.228±.001 |
| | Crossformer | **0.139**±.002 | **0.234**±.003 | 0.141±.001 | 0.235±.001 |
| | iTransformer | **0.154**±.000 | **0.245**±.000 | 0.167±.000 | 0.254±.000 |
| | DLinear | **0.193**±.000 | 0.277±.000 | 0.195±.000 | 0.277±.000 |
| Weather | Transf. | **0.153**±.001 | **0.198**±.000 | 0.161±.000 | 0.208±.001 |
| | PatchTST | **0.174**±.000 | **0.213**±.001 | 0.180±.002 | 0.221±.002 |
| | Crossformer | 0.154±.003 | 0.225±.002 | 0.154±.003 | 0.225±.004 |
| | iTransformer | **0.170**±.001 | **0.211**±.001 | 0.176±.000 | 0.217±.001 |
| | DLinear | 0.199±.005 | 0.258±.008 | **0.196**±.001 | **0.248**±.002 |
| Traffic | Transf. | **0.417**±.009 | **0.278**±.005 | 0.479±.006 | 0.289±.001 |
| | PatchTST | **0.355**±.000 | **0.244**±.000 | 0.383±.001 | 0.261±.001 |
| | Crossformer | 0.548±.024 | **0.278**±.011 | **0.540**±.014 | 0.279±.007 |
| | iTransformer | **0.409**±.000 | **0.277**±.001 | 0.444±.001 | 0.290±.001 |
| | DLinear | **0.609**±.000 | **0.391**±.000 | 0.648±.000 | 0.395±.000 |
| Solar | Transf. | **0.196**±.000 | **0.243**±.001 | 0.206±.003 | 0.249±.004 |
| | PatchTST | **0.196**±.001 | **0.246**±.004 | 0.225±.003 | 0.268±.003 |
| | Crossformer | 0.176±.006 | 0.231±.010 | 0.177±.008 | **0.215**±.004 |
| | iTransformer | **0.197**±.002 | **0.243**±.001 | 0.221±.003 | 0.256±.002 |
| | DLinear | **0.246**±.001 | **0.331**±.000 | 0.285±.001 | 0.372±.001 |

usage, on the Electricity dataset for a forecasting horizon of 96.

### E.4. Empirical Setup and Additional Experiments for D4: Spatial Processing

Analogously to Sec. 3.3, results in Sec. 3.4 were obtained through extensive hyperparameter tuning for each model, configured with exogenous inputs as hybrid global-local embeddings. Since spatial processing often increases computational cost and reduces memory efficiency, we restrict the input window size to 96. The results in Tab. 12 extend Tab. 4a to include both MSE and MAE, and compare the performance of the reference architectures against baselines that include spatial processing. The reference architectures incorporate either a MLP or a pyramidal attention module for temporal processing, followed by a spatial attention module. The MLP and pyramidal attention were chosen for their advantageous trade-off between performance and computational efficiency (see Fig. 4 and Tab. 11). The ablation study in Tab. 4a compares iTransformer against its variant that replaces space attention with a simple feedforward layer. Tab. 13 –Tab. 16 expand the iTransformer ablation study to additional window sizes (96, 336, 720) and forecasting horizons (96, 336, 720) for both MSE and MAE. These additional experiments further reinforce our analysis in Sec. 3.4, showing a general performance improvement when removing one of the core components of a SOTA architectures. Analogously to Tab. 11 and Fig. 4, Tab. 17 and Fig. 5 show the trade-off between model performance and computational efficiency on the Electricity dataset for a forecasting horizon of 96.

## F. Implementation Details

Our code is implemented in Python (Van Rossum et al., 1995), with the use of the following libraries:

- PyTorch (Paszke et al., 2019);

- PyTorch Geometric (Fey and Lenssen, 2019);

- Torch Spatiotemporal (Cini and Marisca, 2022);

*Table 10.* Results (MSE and MAE) for multiple horizons (H). Best mean results are in **bold**, second best are underlined.

| Model | H | Electricity | | Weather | | Traffic | | Solar | |
|---|---|---|---|---|---|---|---|---|---|
| | | MSE | MAE | MSE | MAE | MSE | MAE | MSE | MAE |
| OLS Global | 96 | 0.140 | 0.237 | 0.174 | 0.234 | 0.410 | 0.282 | 0.222 | 0.291 |
| | 192 | 0.154 | 0.250 | 0.215 | 0.272 | 0.423 | 0.288 | 0.249 | 0.309 |
| | 336 | 0.169 | 0.268 | 0.260 | 0.309 | 0.436 | 0.295 | 0.269 | 0.324 |
| | 720 | 0.204 | 0.301 | 0.323 | 0.361 | 0.466 | 0.315 | 0.271 | 0.327 |
| OLS Local | 96 | 0.134 | 0.230 | **0.144** | 0.209 | 0.426 | 0.298 | 0.223 | 0.295 |
| | 192 | 0.149 | 0.245 | **0.187** | 0.254 | 0.438 | 0.304 | 0.251 | 0.313 |
| | 336 | 0.165 | 0.263 | **0.240** | 0.298 | 0.452 | 0.312 | 0.270 | 0.327 |
| | 720 | 0.201 | 0.297 | **0.316** | 0.358 | 0.482 | 0.330 | 0.272 | 0.331 |
| R. MLP | 96 | 0.129$_{\pm0.000}$ | 0.225$_{\pm0.000}$ | 0.148$_{\pm0.001}$ | 0.198$_{\pm0.000}$ | 0.376$_{\pm0.000}$ | 0.253$_{\pm0.001}$ | 0.194$_{\pm0.003}$ | 0.239$_{\pm0.002}$ |
| | 192 | 0.149$_{\pm0.000}$ | 0.245$_{\pm0.001}$ | 0.191$_{\pm0.001}$ | 0.241$_{\pm0.000}$ | 0.403$_{\pm0.001}$ | 0.270$_{\pm0.001}$ | 0.227$_{\pm0.001}$ | 0.262$_{\pm0.001}$ |
| | 336 | 0.166$_{\pm0.000}$ | 0.262$_{\pm0.000}$ | 0.245$_{\pm0.001}$ | 0.281$_{\pm0.001}$ | 0.417$_{\pm0.001}$ | 0.277$_{\pm0.001}$ | 0.249$_{\pm0.001}$ | 0.277$_{\pm0.002}$ |
| | 720 | 0.205$_{\pm0.000}$ | 0.296$_{\pm0.000}$ | 0.324$_{\pm0.002}$ | 0.338$_{\pm0.001}$ | 0.456$_{\pm0.001}$ | 0.295$_{\pm0.001}$ | 0.254$_{\pm0.000}$ | 0.279$_{\pm0.001}$ |
| R. RNN | 96 | 0.127$_{\pm0.001}$ | 0.221$_{\pm0.000}$ | 0.148$_{\pm0.000}$ | 0.199$_{\pm0.001}$ | 0.362$_{\pm0.002}$ | 0.248$_{\pm0.001}$ | 0.192$_{\pm0.002}$ | **0.236**$_{\pm0.001}$ |
| | 192 | 0.167$_{\pm0.001}$ | 0.267$_{\pm0.001}$ | 0.190$_{\pm0.001}$ | 0.244$_{\pm0.001}$ | 0.411$_{\pm0.005}$ | 0.282$_{\pm0.005}$ | 0.230$_{\pm0.003}$ | 0.270$_{\pm0.002}$ |
| | 336 | 0.186$_{\pm0.001}$ | 0.287$_{\pm0.001}$ | 0.244$_{\pm0.001}$ | 0.286$_{\pm0.001}$ | 0.423$_{\pm0.019}$ | 0.285$_{\pm0.005}$ | 0.251$_{\pm0.007}$ | 0.284$_{\pm0.007}$ |
| | 720 | 0.225$_{\pm0.002}$ | 0.323$_{\pm0.001}$ | 0.324$_{\pm0.003}$ | 0.342$_{\pm0.003}$ | 0.497$_{\pm0.040}$ | 0.297$_{\pm0.001}$ | 0.249$_{\pm0.005}$ | 0.281$_{\pm0.005}$ |
| R. TCN | 96 | 0.130$_{\pm0.000}$ | 0.224$_{\pm0.000}$ | 0.148$_{\pm0.000}$ | 0.200$_{\pm0.001}$ | 0.364$_{\pm0.003}$ | 0.253$_{\pm0.002}$ | 0.193$_{\pm0.004}$ | 0.243$_{\pm0.005}$ |
| | 192 | 0.148$_{\pm0.000}$ | 0.240$_{\pm0.000}$ | 0.195$_{\pm0.001}$ | 0.246$_{\pm0.000}$ | 0.382$_{\pm0.001}$ | 0.261$_{\pm0.001}$ | **0.221**$_{\pm0.002}$ | **0.253**$_{\pm0.002}$ |
| | 336 | 0.165$_{\pm0.001}$ | 0.258$_{\pm0.001}$ | 0.252$_{\pm0.002}$ | 0.289$_{\pm0.001}$ | 0.397$_{\pm0.002}$ | 0.274$_{\pm0.005}$ | 0.249$_{\pm0.005}$ | **0.271**$_{\pm0.005}$ |
| | 720 | 0.202$_{\pm0.002}$ | 0.292$_{\pm0.001}$ | 0.329$_{\pm0.001}$ | 0.342$_{\pm0.001}$ | **0.434**$_{\pm0.001}$ | 0.291$_{\pm0.005}$ | 0.246$_{\pm0.004}$ | 0.273$_{\pm0.003}$ |
| R. Transf. | 96 | 0.129$_{\pm0.001}$ | 0.222$_{\pm0.001}$ | 0.149$_{\pm0.001}$ | 0.203$_{\pm0.002}$ | 0.362$_{\pm0.003}$ | 0.249$_{\pm0.002}$ | 0.203$_{\pm0.006}$ | 0.245$_{\pm0.002}$ |
| | 192 | 0.146$_{\pm0.000}$ | 0.238$_{\pm0.000}$ | 0.198$_{\pm0.001}$ | 0.250$_{\pm0.002}$ | **0.374**$_{\pm0.001}$ | **0.255**$_{\pm0.001}$ | 0.224$_{\pm0.001}$ | 0.260$_{\pm0.001}$ |
| | 336 | 0.164$_{\pm0.001}$ | 0.258$_{\pm0.001}$ | 0.248$_{\pm0.001}$ | 0.289$_{\pm0.002}$ | **0.393**$_{\pm0.002}$ | 0.271$_{\pm0.007}$ | 0.245$_{\pm0.004}$ | **0.271**$_{\pm0.003}$ |
| | 720 | 0.202$_{\pm0.003}$ | 0.294$_{\pm0.002}$ | 0.330$_{\pm0.010}$ | 0.344$_{\pm0.006}$ | **0.434**$_{\pm0.004}$ | 0.294$_{\pm0.008}$ | 0.248$_{\pm0.004}$ | 0.277$_{\pm0.004}$ |
| R. Pyraf. | 96 | 0.129$_{\pm0.001}$ | 0.224$_{\pm0.001}$ | 0.148$_{\pm0.001}$ | 0.199$_{\pm0.001}$ | 0.365$_{\pm0.002}$ | 0.251$_{\pm0.003}$ | **0.189**$_{\pm0.003}$ | **0.236**$_{\pm0.004}$ |
| | 192 | 0.147$_{\pm0.001}$ | 0.240$_{\pm0.000}$ | 0.196$_{\pm0.003}$ | 0.246$_{\pm0.002}$ | 0.384$_{\pm0.003}$ | 0.262$_{\pm0.004}$ | 0.224$_{\pm0.001}$ | 0.256$_{\pm0.001}$ |
| | 336 | 0.164$_{\pm0.001}$ | 0.258$_{\pm0.000}$ | 0.248$_{\pm0.003}$ | 0.287$_{\pm0.003}$ | 0.397$_{\pm0.002}$ | 0.269$_{\pm0.002}$ | **0.244**$_{\pm0.001}$ | **0.271**$_{\pm0.002}$ |
| | 720 | 0.200$_{\pm0.000}$ | 0.293$_{\pm0.000}$ | 0.328$_{\pm0.003}$ | 0.344$_{\pm0.002}$ | **0.434**$_{\pm0.002}$ | 0.297$_{\pm0.003}$ | 0.248$_{\pm0.002}$ | **0.272**$_{\pm0.001}$ |
| TimeMixer | 96 | 0.129$_{\pm0.001}$ | 0.224$_{\pm0.000}$ | 0.147$_{\pm0.001}$ | 0.197$_{\pm0.000}$ | 0.373$_{\pm0.002}$ | 0.271$_{\pm0.003}$ | 0.199$_{\pm0.001}$ | 0.245$_{\pm0.000}$ |
| | 192 | 0.147$_{\pm0.001}$ | 0.241$_{\pm0.000}$ | 0.191$_{\pm0.000}$ | **0.239**$_{\pm0.000}$ | 0.396$_{\pm0.001}$ | 0.283$_{\pm0.001}$ | 0.230$_{\pm0.003}$ | 0.268$_{\pm0.003}$ |
| | 336 | 0.166$_{\pm0.001}$ | 0.260$_{\pm0.001}$ | 0.244$_{\pm0.002}$ | 0.281$_{\pm0.002}$ | 0.416$_{\pm0.001}$ | 0.297$_{\pm0.001}$ | **0.244**$_{\pm0.002}$ | 0.280$_{\pm0.000}$ |
| | 720 | 0.206$_{\pm0.003}$ | 0.297$_{\pm0.003}$ | 0.321$_{\pm0.003}$ | 0.334$_{\pm0.003}$ | 0.450$_{\pm0.006}$ | 0.318$_{\pm0.003}$ | **0.245**$_{\pm0.004}$ | 0.280$_{\pm0.001}$ |
| PatchTST | 96 | **0.125**$_{\pm0.000}$ | **0.218**$_{\pm0.000}$ | 0.148$_{\pm0.001}$ | **0.195**$_{\pm0.001}$ | **0.345**$_{\pm0.000}$ | **0.234**$_{\pm0.000}$ | 0.197$_{\pm0.001}$ | 0.244$_{\pm0.004}$ |
| | 192 | **0.143**$_{\pm0.000}$ | **0.236**$_{\pm0.000}$ | 0.194$_{\pm0.000}$ | **0.239**$_{\pm0.001}$ | 0.384$_{\pm0.001}$ | 0.258$_{\pm0.001}$ | 0.227$_{\pm0.002}$ | 0.260$_{\pm0.002}$ |
| | 336 | **0.160**$_{\pm0.000}$ | **0.254**$_{\pm0.000}$ | 0.247$_{\pm0.000}$ | **0.279**$_{\pm0.001}$ | 0.396$_{\pm0.001}$ | **0.264**$_{\pm0.000}$ | 0.249$_{\pm0.001}$ | 0.273$_{\pm0.002}$ |
| | 720 | **0.197**$_{\pm0.001}$ | **0.288**$_{\pm0.001}$ | 0.322$_{\pm0.002}$ | **0.333**$_{\pm0.001}$ | 0.435$_{\pm0.001}$ | **0.286**$_{\pm0.000}$ | 0.247$_{\pm0.001}$ | 0.273$_{\pm0.002}$ |
| DLinear | 96 | 0.140$_{\pm0.000}$ | 0.237$_{\pm0.000}$ | 0.173$_{\pm0.000}$ | 0.232$_{\pm0.001}$ | 0.407$_{\pm0.000}$ | 0.283$_{\pm0.000}$ | 0.246$_{\pm0.001}$ | 0.331$_{\pm0.000}$ |
| | 192 | 0.154$_{\pm0.000}$ | 0.250$_{\pm0.001}$ | 0.216$_{\pm0.001}$ | 0.274$_{\pm0.003}$ | 0.421$_{\pm0.000}$ | 0.290$_{\pm0.000}$ | 0.267$_{\pm0.001}$ | 0.342$_{\pm0.000}$ |
| | 336 | 0.169$_{\pm0.000}$ | 0.268$_{\pm0.000}$ | 0.265$_{\pm0.002}$ | 0.318$_{\pm0.003}$ | 0.433$_{\pm0.000}$ | 0.296$_{\pm0.000}$ | 0.289$_{\pm0.001}$ | 0.353$_{\pm0.000}$ |
| | 720 | 0.203$_{\pm0.000}$ | 0.300$_{\pm0.000}$ | 0.331$_{\pm0.002}$ | 0.373$_{\pm0.003}$ | 0.461$_{\pm0.000}$ | 0.314$_{\pm0.000}$ | 0.294$_{\pm0.001}$ | 0.355$_{\pm0.000}$ |

*Table 11.* Performance and resource utilization of the models selected in 3.3 on the Electricity dataset. Best performance is shown in **bold**, second best is underlined.

| Model | Horizon | Batch Time (ms) | Batches per Second | GPU Mem. (MB) | CUDA Time (ms) |
|---|---|---|---|---|---|
| MLP | 96 | $27.6_{\pm1.4}$ | $36.3_{\pm1.2}$ | 628.0 | 20.3 |
| | 192 | $27.6_{\pm1.4}$ | $36.3_{\pm1.2}$ | 628.0 | 27.1 |
| | 336 | $27.6_{\pm1.4}$ | $36.3_{\pm1.2}$ | 653.2 | 26.3 |
| | 720 | $27.6_{\pm1.4}$ | $36.3_{\pm1.2}$ | 705.6 | 31.8 |
| RNN | 96 | $36.7_{\pm0.9}$ | $28.2_{\pm0.6}$ | 3635.2 | 235.6 |
| | 192 | $37.3_{\pm2.9}$ | $27.8_{\pm1.8}$ | 3643.5 | 243.0 |
| | 336 | $37.3_{\pm2.9}$ | $27.8_{\pm1.8}$ | 3854.3 | 246.7 |
| | 720 | $37.3_{\pm2.9}$ | $27.8_{\pm1.8}$ | 3860.6 | 258.5 |
| TCN | 96 | $29.9_{\pm1.4}$ | $33.5_{\pm1.1}$ | 1217.3 | 62.0 |
| | 192 | $29.9_{\pm1.4}$ | $33.5_{\pm1.1}$ | 1217.3 | 82.0 |
| | 336 | $29.9_{\pm1.4}$ | $33.5_{\pm1.1}$ | 1219.4 | 85.1 |
| | 720 | $29.9_{\pm1.4}$ | $33.5_{\pm1.1}$ | 1240.4 | 87.8 |
| Transf. | 96 | $36.7_{\pm1.3}$ | $27.3_{\pm0.7}$ | 1942.9 | 119.3 |
| | 192 | $36.7_{\pm1.3}$ | $27.3_{\pm0.7}$ | 1961.7 | 144.0 |
| | 336 | $36.7_{\pm1.3}$ | $27.3_{\pm0.7}$ | 1959.7 | 148.1 |
| | 720 | $36.7_{\pm1.3}$ | $27.3_{\pm0.7}$ | 1976.4 | 164.7 |
| Pyraf. | 96 | $41.7_{\pm0.8}$ | $24.0_{\pm0.4}$ | 2559.4 | 189.9 |
| | 192 | $41.7_{\pm0.8}$ | $24.0_{\pm0.4}$ | 2561.5 | 191.9 |
| | 336 | $41.7_{\pm0.8}$ | $24.0_{\pm0.4}$ | 2563.6 | 192.8 |
| | 720 | $41.7_{\pm0.8}$ | $24.0_{\pm0.4}$ | 2567.8 | 198.7 |
| DLinear | 96 | $\mathbf{18.9_{\pm1.1}}$ | $\mathbf{52.9_{\pm1.7}}$ | **615.5** | **10.7** |
| | 192 | $\mathbf{18.9_{\pm1.1}}$ | $\mathbf{52.9_{\pm1.7}}$ | **615.5** | **17.2** |
| | 336 | $\mathbf{18.9_{\pm1.1}}$ | $\mathbf{52.9_{\pm1.7}}$ | **638.5** | **19.1** |
| | 720 | $\mathbf{18.9_{\pm1.1}}$ | $\mathbf{52.9_{\pm1.7}}$ | **699.4** | **22.5** |
| PatchTST | 96 | $34.3_{\pm0.5}$ | $29.1_{\pm0.4}$ | 1445.9 | 74.9 |
| | 192 | $34.3_{\pm0.5}$ | $29.1_{\pm0.4}$ | 1443.8 | 94.6 |
| | 336 | $34.3_{\pm0.5}$ | $29.1_{\pm0.4}$ | 1443.8 | 93.6 |
| | 720 | $34.3_{\pm0.5}$ | $29.1_{\pm0.4}$ | 1462.7 | 107.7 |
| TimeMixer | 96 | $97.6_{\pm93.5}$ | $10.9_{\pm0.7}$ | 2301.5 | 410.4 |
| | 192 | $97.6_{\pm93.5}$ | $10.9_{\pm0.7}$ | 2303.6 | 405.1 |
| | 336 | $97.6_{\pm93.5}$ | $10.9_{\pm0.7}$ | 2311.9 | 375.9 |
| | 720 | $97.6_{\pm93.5}$ | $10.9_{\pm0.7}$ | 2450.3 | 478.8 |

*Table 12.* Results (MSE and MAE) across datasets and horizons (H). Best mean results are in **bold**, second best are underlined. A dash (–) marks experiments beyond our computational budget.

| Model | H | Electricity | | Weather | | Traffic | | Solar | |
|---|---|---|---|---|---|---|---|---|---|
| | | MSE | MAE | MSE | MAE | MSE | MAE | MSE | MAE |
| MLP + sp. attn. | 96 | $0.140_{\pm0.001}$ | $0.238_{\pm0.001}$ | $0.157_{\pm0.000}$ | $0.202_{\pm0.001}$ | $0.435_{\pm0.006}$ | $0.275_{\pm0.001}$ | $0.201_{\pm0.009}$ | $0.246_{\pm0.003}$ |
| | 192 | $0.167_{\pm0.002}$ | $0.263_{\pm0.002}$ | $0.206_{\pm0.001}$ | $\mathbf{0.249_{\pm0.001}}$ | $0.454_{\pm0.008}$ | $0.286_{\pm0.003}$ | $0.237_{\pm0.003}$ | $0.272_{\pm0.003}$ |
| | 336 | $0.182_{\pm0.002}$ | $0.281_{\pm0.002}$ | $0.264_{\pm0.001}$ | $0.292_{\pm0.001}$ | $0.472_{\pm0.007}$ | $0.292_{\pm0.004}$ | $0.266_{\pm0.006}$ | $0.294_{\pm0.005}$ |
| | 720 | $0.209_{\pm0.008}$ | $0.305_{\pm0.006}$ | $0.349_{\pm0.001}$ | $0.345_{\pm0.001}$ | $0.523_{\pm0.003}$ | $0.316_{\pm0.003}$ | $0.266_{\pm0.004}$ | $0.304_{\pm0.007}$ |
| Pyraf. + sp. attn. | 96 | $0.139_{\pm0.001}$ | $0.236_{\pm0.001}$ | $0.157_{\pm0.002}$ | $0.204_{\pm0.001}$ | $\mathbf{0.389_{\pm0.002}}$ | $0.267_{\pm0.001}$ | $0.188_{\pm0.002}$ | $0.235_{\pm0.003}$ |
| | 192 | $0.157_{\pm0.000}$ | $0.254_{\pm0.000}$ | $0.207_{\pm0.001}$ | $0.251_{\pm0.002}$ | $\mathbf{0.410_{\pm0.001}}$ | $\mathbf{0.276_{\pm0.002}}$ | $0.234_{\pm0.003}$ | $0.270_{\pm0.008}$ |
| | 336 | $0.174_{\pm0.002}$ | $0.273_{\pm0.002}$ | $0.267_{\pm0.001}$ | $0.294_{\pm0.002}$ | $\mathbf{0.420_{\pm0.001}}$ | $\mathbf{0.283_{\pm0.001}}$ | $0.248_{\pm0.001}$ | $0.281_{\pm0.004}$ |
| | 720 | $\mathbf{0.194_{\pm0.001}}$ | $\mathbf{0.292_{\pm0.001}}$ | $0.348_{\pm0.002}$ | $0.347_{\pm0.000}$ | $\mathbf{0.449_{\pm0.002}}$ | $\mathbf{0.300_{\pm0.001}}$ | $0.249_{\pm0.001}$ | $0.280_{\pm0.002}$ |
| iTransformer | 96 | $0.148_{\pm0.000}$ | $0.241_{\pm0.000}$ | $0.171_{\pm0.001}$ | $0.210_{\pm0.001}$ | $0.393_{\pm0.001}$ | $\mathbf{0.266_{\pm0.001}}$ | $0.208_{\pm0.003}$ | $0.240_{\pm0.006}$ |
| | 192 | $0.166_{\pm0.001}$ | $0.259_{\pm0.001}$ | $0.221_{\pm0.001}$ | $0.255_{\pm0.000}$ | $0.424_{\pm0.001}$ | $0.286_{\pm0.001}$ | $0.233_{\pm0.003}$ | $0.257_{\pm0.003}$ |
| | 336 | $0.179_{\pm0.002}$ | $0.274_{\pm0.002}$ | $0.276_{\pm0.000}$ | $0.295_{\pm0.000}$ | $0.430_{\pm0.001}$ | $0.289_{\pm0.001}$ | $0.244_{\pm0.000}$ | $0.273_{\pm0.002}$ |
| | 720 | $0.218_{\pm0.000}$ | $0.309_{\pm0.001}$ | $0.353_{\pm0.000}$ | $0.346_{\pm0.000}$ | $0.455_{\pm0.001}$ | $0.303_{\pm0.000}$ | $0.255_{\pm0.002}$ | $0.280_{\pm0.001}$ |
| Crosformer | 96 | $\mathbf{0.136_{\pm0.000}}$ | $\mathbf{0.232_{\pm0.001}}$ | $\mathbf{0.152_{\pm0.003}}$ | $0.222_{\pm0.004}$ | $0.527_{\pm0.002}$ | $0.270_{\pm0.003}$ | $\mathbf{0.184_{\pm0.008}}$ | $0.227_{\pm0.006}$ |
| | 192 | $0.161_{\pm0.003}$ | $0.257_{\pm0.003}$ | $\mathbf{0.200_{\pm0.007}}$ | $0.271_{\pm0.007}$ | $0.541_{\pm0.002}$ | $\mathbf{0.276_{\pm0.002}}$ | $\mathbf{0.207_{\pm0.015}}$ | $\mathbf{0.246_{\pm0.003}}$ |
| | 336 | $0.187_{\pm0.007}$ | $0.286_{\pm0.007}$ | $0.263_{\pm0.001}$ | $0.325_{\pm0.003}$ | $0.561_{\pm0.005}$ | $0.289_{\pm0.002}$ | $\mathbf{0.222_{\pm0.016}}$ | $\mathbf{0.252_{\pm0.001}}$ |
| | 720 | – | – | $0.359_{\pm0.007}$ | $0.389_{\pm0.004}$ | – | – | $\mathbf{0.214_{\pm0.001}}$ | $\mathbf{0.250_{\pm0.004}}$ |
| ModernTCN | 96 | $0.141_{\pm0.000}$ | $0.237_{\pm0.001}$ | $0.154_{\pm0.001}$ | $\mathbf{0.200_{\pm0.001}}$ | $0.445_{\pm0.001}$ | $0.287_{\pm0.001}$ | $0.190_{\pm0.001}$ | $\mathbf{0.222_{\pm0.002}}$ |
| | 192 | $\mathbf{0.156_{\pm0.001}}$ | $\mathbf{0.249_{\pm0.001}}$ | $0.201_{\pm0.002}$ | $0.252_{\pm0.002}$ | – | – | $0.235_{\pm0.001}$ | $0.250_{\pm0.002}$ |
| | 336 | $\mathbf{0.172_{\pm0.003}}$ | $\mathbf{0.263_{\pm0.001}}$ | $\mathbf{0.254_{\pm0.003}}$ | $\mathbf{0.291_{\pm0.002}}$ | – | – | $0.258_{\pm0.004}$ | $0.266_{\pm0.003}$ |
| | 720 | $0.201_{\pm0.000}$ | $\mathbf{0.292_{\pm0.000}}$ | $\mathbf{0.338_{\pm0.005}}$ | $\mathbf{0.343_{\pm0.005}}$ | – | – | $0.262_{\pm0.003}$ | $0.277_{\pm0.004}$ |

*Table 13.* Ablation study on iTransformer for window=96 across different forecasting horizons. Best mean results are in **bold**.

| Dataset | Horizon | With space attention | | Without space attention | |
|---|---|---|---|---|---|
| | | MSE | MAE | MSE | MAE |
| Electricity | 96 | $\mathbf{0.148_{\pm0.000}}$ | $0.241_{\pm0.000}$ | $0.149_{\pm0.001}$ | $\mathbf{0.237_{\pm0.001}}$ |
| | 192 | $0.166_{\pm0.001}$ | $0.259_{\pm0.001}$ | $\mathbf{0.161_{\pm0.000}}$ | $\mathbf{0.250_{\pm0.000}}$ |
| | 336 | $0.179_{\pm0.002}$ | $0.274_{\pm0.002}$ | $0.179_{\pm0.000}$ | $\mathbf{0.268_{\pm0.000}}$ |
| | 720 | $\mathbf{0.218_{\pm0.000}}$ | $0.309_{\pm0.001}$ | $0.219_{\pm0.000}$ | $\mathbf{0.303_{\pm0.000}}$ |
| Weather | 96 | $0.171_{\pm0.001}$ | $0.210_{\pm0.001}$ | $0.171_{\pm0.000}$ | $0.210_{\pm0.001}$ |
| | 192 | $0.221_{\pm0.001}$ | $0.255_{\pm0.000}$ | $\mathbf{0.219_{\pm0.001}}$ | $\mathbf{0.253_{\pm0.001}}$ |
| | 336 | $0.276_{\pm0.000}$ | $0.295_{\pm0.000}$ | $\mathbf{0.275_{\pm0.000}}$ | $\mathbf{0.294_{\pm0.000}}$ |
| | 720 | $0.353_{\pm0.000}$ | $0.346_{\pm0.000}$ | $\mathbf{0.352_{\pm0.001}}$ | $\mathbf{0.345_{\pm0.000}}$ |
| Traffic | 96 | $0.393_{\pm0.001}$ | $0.266_{\pm0.001}$ | $\mathbf{0.390_{\pm0.001}}$ | $\mathbf{0.258_{\pm0.000}}$ |
| | 192 | $0.424_{\pm0.001}$ | $0.286_{\pm0.001}$ | $\mathbf{0.409_{\pm0.000}}$ | $\mathbf{0.268_{\pm0.000}}$ |
| | 336 | $0.430_{\pm0.001}$ | $0.289_{\pm0.001}$ | $\mathbf{0.423_{\pm0.000}}$ | $\mathbf{0.274_{\pm0.000}}$ |
| | 720 | $0.455_{\pm0.001}$ | $0.303_{\pm0.000}$ | $\mathbf{0.454_{\pm0.000}}$ | $\mathbf{0.292_{\pm0.000}}$ |
| Solar | 96 | $0.208_{\pm0.003}$ | $0.240_{\pm0.006}$ | $\mathbf{0.194_{\pm0.001}}$ | $\mathbf{0.230_{\pm0.002}}$ |
| | 192 | $0.233_{\pm0.003}$ | $0.257_{\pm0.003}$ | $\mathbf{0.226_{\pm0.001}}$ | $0.257_{\pm0.003}$ |
| | 336 | $\mathbf{0.244_{\pm0.000}}$ | $0.273_{\pm0.002}$ | $0.245_{\pm0.003}$ | $\mathbf{0.267_{\pm0.002}}$ |
| | 720 | $0.255_{\pm0.002}$ | $0.280_{\pm0.001}$ | $\mathbf{0.249_{\pm0.001}}$ | $\mathbf{0.271_{\pm0.001}}$ |

*Table 14.* Ablation study on iTransformer for window=336 across different forecasting horizons. Best mean results are in **bold**.

| Dataset | Horizon | With space attention | | Without space attention | |
|---|---|---|---|---|---|
| | | MSE | MAE | MSE | MAE |
| Electricity | 96 | $0.135_{\pm0.001}$ | $0.229_{\pm0.001}$ | $\mathbf{0.130}_{\pm\mathbf{0.000}}$ | $\mathbf{0.223}_{\pm\mathbf{0.000}}$ |
| | 192 | $0.155_{\pm0.001}$ | $0.248_{\pm0.001}$ | $\mathbf{0.149}_{\pm\mathbf{0.000}}$ | $\mathbf{0.242}_{\pm\mathbf{0.000}}$ |
| | 336 | $0.172_{\pm0.002}$ | $0.267_{\pm0.000}$ | $\mathbf{0.166}_{\pm\mathbf{0.000}}$ | $\mathbf{0.260}_{\pm\mathbf{0.000}}$ |
| | 720 | $\mathbf{0.201}_{\pm\mathbf{0.002}}$ | $0.294_{\pm0.002}$ | $0.205_{\pm0.000}$ | $0.294_{\pm0.000}$ |
| Weather | 96 | $0.159_{\pm0.001}$ | $0.207_{\pm0.000}$ | $\mathbf{0.153}_{\pm\mathbf{0.000}}$ | $\mathbf{0.202}_{\pm\mathbf{0.001}}$ |
| | 192 | $0.203_{\pm0.001}$ | $0.249_{\pm0.001}$ | $\mathbf{0.197}_{\pm\mathbf{0.001}}$ | $\mathbf{0.245}_{\pm\mathbf{0.002}}$ |
| | 336 | $0.252_{\pm0.002}$ | $0.286_{\pm0.000}$ | $\mathbf{0.249}_{\pm\mathbf{0.001}}$ | $\mathbf{0.284}_{\pm\mathbf{0.001}}$ |
| | 720 | $\mathbf{0.326}_{\pm\mathbf{0.003}}$ | $\mathbf{0.338}_{\pm\mathbf{0.002}}$ | $0.328_{\pm0.002}$ | $0.339_{\pm0.001}$ |
| Traffic | 96 | $0.363_{\pm0.000}$ | $0.257_{\pm0.001}$ | $\mathbf{0.359}_{\pm\mathbf{0.001}}$ | $\mathbf{0.247}_{\pm\mathbf{0.001}}$ |
| | 192 | $0.385_{\pm0.002}$ | $0.269_{\pm0.001}$ | $\mathbf{0.377}_{\pm\mathbf{0.001}}$ | $\mathbf{0.256}_{\pm\mathbf{0.001}}$ |
| | 336 | $0.397_{\pm0.002}$ | $0.277_{\pm0.001}$ | $\mathbf{0.386}_{\pm\mathbf{0.000}}$ | $\mathbf{0.261}_{\pm\mathbf{0.000}}$ |
| | 720 | $0.423_{\pm0.001}$ | $0.291_{\pm0.000}$ | $\mathbf{0.420}_{\pm\mathbf{0.001}}$ | $\mathbf{0.282}_{\pm\mathbf{0.000}}$ |
| Solar | 96 | $0.195_{\pm0.000}$ | $0.252_{\pm0.003}$ | $\mathbf{0.189}_{\pm\mathbf{0.001}}$ | $\mathbf{0.232}_{\pm\mathbf{0.001}}$ |
| | 192 | $0.222_{\pm0.004}$ | $0.269_{\pm0.001}$ | $\mathbf{0.207}_{\pm\mathbf{0.000}}$ | $\mathbf{0.249}_{\pm\mathbf{0.000}}$ |
| | 336 | $0.230_{\pm0.007}$ | $0.276_{\pm0.004}$ | $\mathbf{0.214}_{\pm\mathbf{0.000}}$ | $\mathbf{0.255}_{\pm\mathbf{0.001}}$ |
| | 720 | $0.223_{\pm0.002}$ | $0.274_{\pm0.003}$ | $\mathbf{0.216}_{\pm\mathbf{0.003}}$ | $\mathbf{0.258}_{\pm\mathbf{0.002}}$ |

*Table 15.* Ablation study on iTransformer for window=720 across different forecasting horizons. Best mean results are in **bold**.

| Dataset | Horizon | With space attention | | Without space attention | |
|---|---|---|---|---|---|
| | | MSE | MAE | MSE | MAE |
| Electricity | 96 | $0.135_{\pm0.002}$ | $0.231_{\pm0.001}$ | $\mathbf{0.132}_{\pm\mathbf{0.000}}$ | $\mathbf{0.227}_{\pm\mathbf{0.001}}$ |
| | 192 | $0.157_{\pm0.001}$ | $0.252_{\pm0.001}$ | $\mathbf{0.151}_{\pm\mathbf{0.001}}$ | $\mathbf{0.245}_{\pm\mathbf{0.001}}$ |
| | 336 | $0.175_{\pm0.001}$ | $0.272_{\pm0.001}$ | $\mathbf{0.165}_{\pm\mathbf{0.000}}$ | $\mathbf{0.263}_{\pm\mathbf{0.000}}$ |
| | 720 | $\mathbf{0.195}_{\pm\mathbf{0.001}}$ | $\mathbf{0.289}_{\pm\mathbf{0.002}}$ | $0.201_{\pm0.001}$ | $0.294_{\pm0.001}$ |
| Weather | 96 | $0.155_{\pm0.002}$ | $0.208_{\pm0.002}$ | $\mathbf{0.149}_{\pm\mathbf{0.000}}$ | $\mathbf{0.201}_{\pm\mathbf{0.001}}$ |
| | 192 | $0.202_{\pm0.002}$ | $0.250_{\pm0.001}$ | $\mathbf{0.195}_{\pm\mathbf{0.001}}$ | $\mathbf{0.247}_{\pm\mathbf{0.002}}$ |
| | 336 | $0.249_{\pm0.002}$ | $\mathbf{0.288}_{\pm\mathbf{0.002}}$ | $0.249_{\pm0.001}$ | $0.289_{\pm0.001}$ |
| | 720 | $0.322_{\pm0.002}$ | $0.342_{\pm0.000}$ | $\mathbf{0.317}_{\pm\mathbf{0.001}}$ | $\mathbf{0.337}_{\pm\mathbf{0.001}}$ |
| Traffic | 96 | $\mathbf{0.353}_{\pm\mathbf{0.004}}$ | $0.256_{\pm0.001}$ | $0.357_{\pm0.000}$ | $\mathbf{0.251}_{\pm\mathbf{0.000}}$ |
| | 192 | $0.372_{\pm0.002}$ | $0.267_{\pm0.000}$ | $\mathbf{0.369}_{\pm\mathbf{0.000}}$ | $\mathbf{0.260}_{\pm\mathbf{0.000}}$ |
| | 336 | $0.388_{\pm0.002}$ | $0.276_{\pm0.001}$ | $\mathbf{0.383}_{\pm\mathbf{0.001}}$ | $\mathbf{0.269}_{\pm\mathbf{0.001}}$ |
| | 720 | $\mathbf{0.417}_{\pm\mathbf{0.002}}$ | $0.290_{\pm0.001}$ | $0.420_{\pm0.000}$ | $\mathbf{0.287}_{\pm\mathbf{0.000}}$ |
| Solar | 96 | $\mathbf{0.175}_{\pm\mathbf{0.003}}$ | $0.245_{\pm0.004}$ | $0.182_{\pm0.008}$ | $\mathbf{0.235}_{\pm\mathbf{0.006}}$ |
| | 192 | $\mathbf{0.197}_{\pm\mathbf{0.001}}$ | $0.262_{\pm0.002}$ | $0.200_{\pm0.003}$ | $\mathbf{0.258}_{\pm\mathbf{0.003}}$ |
| | 336 | $0.211_{\pm0.002}$ | $0.272_{\pm0.003}$ | $\mathbf{0.209}_{\pm\mathbf{0.002}}$ | $\mathbf{0.262}_{\pm\mathbf{0.002}}$ |
| | 720 | $0.216_{\pm0.001}$ | $0.275_{\pm0.003}$ | $\mathbf{0.210}_{\pm\mathbf{0.000}}$ | $\mathbf{0.264}_{\pm\mathbf{0.001}}$ |

*Table 16.* Ablation study on iTransformer for horizon=96 across different window lengths. Best mean results are in **bold**.

| Dataset | Window | With space attention | | Without space attention | |
|---|---|---|---|---|---|
| | | MSE | MAE | MSE | MAE |
| Electricity | 96 | **0.148**$_{\pm0.000}$ | 0.241$_{\pm0.000}$ | 0.149$_{\pm0.001}$ | **0.237**$_{\pm0.001}$ |
| | 336 | 0.135$_{\pm0.001}$ | 0.229$_{\pm0.001}$ | **0.130**$_{\pm0.000}$ | **0.223**$_{\pm0.000}$ |
| | 720 | 0.135$_{\pm0.002}$ | 0.231$_{\pm0.001}$ | **0.132**$_{\pm0.000}$ | **0.227**$_{\pm0.001}$ |
| Weather | 96 | 0.171$_{\pm0.001}$ | 0.210$_{\pm0.001}$ | 0.171$_{\pm0.000}$ | 0.210$_{\pm0.001}$ |
| | 336 | 0.159$_{\pm0.001}$ | 0.207$_{\pm0.000}$ | **0.153**$_{\pm0.000}$ | **0.202**$_{\pm0.001}$ |
| | 720 | 0.155$_{\pm0.002}$ | 0.208$_{\pm0.002}$ | **0.149**$_{\pm0.000}$ | **0.201**$_{\pm0.001}$ |
| Traffic | 96 | 0.393$_{\pm0.001}$ | 0.266$_{\pm0.001}$ | **0.390**$_{\pm0.001}$ | **0.258**$_{\pm0.000}$ |
| | 336 | 0.363$_{\pm0.000}$ | 0.257$_{\pm0.001}$ | **0.359**$_{\pm0.001}$ | **0.247**$_{\pm0.001}$ |
| | 720 | **0.353**$_{\pm0.004}$ | 0.256$_{\pm0.001}$ | 0.357$_{\pm0.000}$ | **0.251**$_{\pm0.000}$ |
| Solar | 96 | 0.208$_{\pm0.003}$ | 0.240$_{\pm0.006}$ | **0.194**$_{\pm0.001}$ | **0.230**$_{\pm0.002}$ |
| | 336 | 0.195$_{\pm0.000}$ | 0.252$_{\pm0.003}$ | **0.189**$_{\pm0.001}$ | **0.232**$_{\pm0.001}$ |
| | 720 | **0.175**$_{\pm0.003}$ | 0.245$_{\pm0.004}$ | 0.182$_{\pm0.008}$ | **0.235**$_{\pm0.006}$ |

*Table 17.* Performance and resource utilization of the models selected in 4a on the Electricity dataset. Best performance is shown in **bold**, second best is underlined.

| Model | Horizon | Batch Time (ms) | Batches per Second | GPU Mem. (MB) | CUDA Time (ms) |
|---|---|---|---|---|---|
| MLP + sp. att. | 96 | **40.2**$_{\pm1.0}$ | **24.9**$_{\pm0.5}$ | **1162.8** | **96.9** |
| | 192 | **40.6**$_{\pm1.3}$ | **24.7**$_{\pm0.6}$ | **1236.2** | **123.8** |
| | 336 | **40.6**$_{\pm1.3}$ | **24.7**$_{\pm0.6}$ | **1215.2** | **155.4** |
| | 720 | **40.6**$_{\pm1.3}$ | **24.7**$_{\pm0.6}$ | **1357.8** | **185.8** |
| Pyraf. + sp. att. | 96 | 96.4$_{\pm0.4}$ | 10.4$_{\pm0.0}$ | 7822.9 | 783.6 |
| | 192 | 96.4$_{\pm0.4}$ | 10.4$_{\pm0.0}$ | 7908.8 | 787.3 |
| | 336 | 96.4$_{\pm0.4}$ | 10.4$_{\pm0.0}$ | 7837.5 | 808.8 |
| | 720 | 96.4$_{\pm0.4}$ | 10.4$_{\pm0.0}$ | 7940.3 | 861.9 |
| iTransformer | 96 | 50.8$_{\pm1.3}$ | 19.7$_{\pm0.4}$ | 1729.0 | 217.4 |
| | 192 | 50.7$_{\pm1.1}$ | 19.7$_{\pm0.4}$ | 1630.4 | 227.9 |
| | 336 | 50.7$_{\pm1.1}$ | 19.7$_{\pm0.4}$ | 1712.2 | 252.9 |
| | 720 | 50.7$_{\pm1.1}$ | 19.7$_{\pm0.4}$ | 1871.6 | 317.8 |
| Crossformer | 96 | 138.0$_{\pm1.0}$ | 7.2$_{\pm0.1}$ | 5702.8 | 912.8 |
| | 192 | 161.9$_{\pm28.1}$ | 6.4$_{\pm1.0}$ | 9412.4 | 1532.0 |
| | 336 | 161.9$_{\pm28.1}$ | 6.4$_{\pm1.0}$ | 16032.6 | 2516.0 |
| | 720 | 161.9$_{\pm28.1}$ | 6.4$_{\pm1.0}$ | 39527.4 | 5589.0 |
| Modern TCN | 96 | 145.4$_{\pm1.1}$ | 6.9$_{\pm0.0}$ | 10620.3 | 1967.0 |
| | 192 | 147.4$_{\pm8.8}$ | 6.8$_{\pm0.3}$ | 10620.3 | 1971.0 |
| | 336 | 147.4$_{\pm8.8}$ | 6.8$_{\pm0.3}$ | 10662.2 | 2005.0 |
| | 720 | 147.4$_{\pm8.8}$ | 6.8$_{\pm0.3}$ | 10857.2 | 2036.0 |

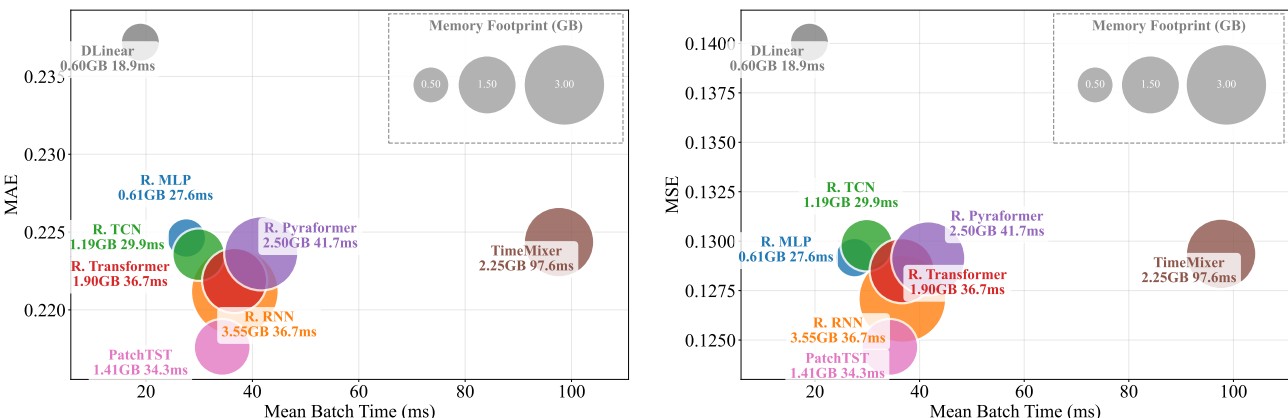

*Figure 4.* MAE and MSE performance versus mean batch time during training for models *not including* spatial processing, on the Electricity dataset for a forecasting horizon of 96 and a batch size of 512. Circle size indicates memory consumption.

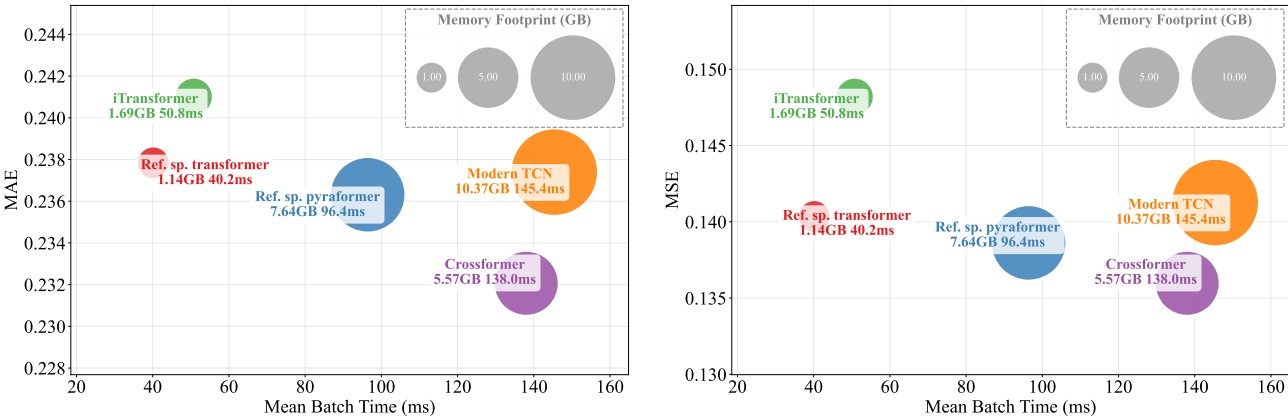

*Figure 5.* MAE and MSE performance versus mean batch time during training for models *including* spatial processing, on the Electricity dataset for a forecasting horizon of 96 and a batch size of 32. Circle size indicates memory consumption.

- Scikit-learn (Pedregosa et al., 2011);

- PyTorch Lightning (Falcon and The PyTorch Lightning team, 2019);

- Hydra (Yadan, 2019);

- Numpy (Harris et al., 2020).

## G. Forecasting Model Card

In Fig. 6, we report an example of the usage of the newly introduced forecasting model card for PatchTST.

## H. Time Series Foundation Models

The benchmarking issues discussed throughout this work also directly apply to time series foundation models. Indeed, misleading benchmarking practices may still lead to incorrect conclusions regarding the effectiveness of specific design choices. In this work, our empirical analysis focuses on forecasting models that can be retrained under controlled configurations, allowing us to isolate the effect of individual design dimensions. Extending the same type of analysis to foundation models would require a fundamentally different experimental setup. Here, we report only an experiment probing the use of exogenous variables (D2), as this is the only design dimension that can be investigated without retraining the foundation model from scratch. As reference architecture, we use Chronos-2 (Ansari et al., 2025). In this experiment, the Electricity dataset is excluded because it was used during Chronos-2 pretraining. Instead, we consider four additional ETT

*Figure 6.* Example of model cards for PatchTST on the Electricity dataset

---

## FORECASTING MODEL CARD

**Model setting**

- *Window length*: fixed lookback window of 336

- *Transductive or inductive (cold start)*: inductive

- *Masking*: not applied/needed

**D1. Model configuration**

- *Global/local/hybrid*: global model

- *Hybrid parameters (non-shared)*: not applicable

**D2. Preprocessing and exogenous variables**

- *Scaling*: standard normalization (z-score) applied per series and in-batch RevInv normalization

- *Covariates/exogenous variables*: not used

**D3. Temporal processing**

- *Temporal modules*: convolutional encoding followed by patching-based Transformer layers

- *Complexity scaling with steps*: the time and space complexity scales quadratically with the number of patches (self-attention)

**D4. Spatial processing**

- *Spatial modules*: not applicable

- *Complexity scaling with nodes*: not applicable

---

(Electricity Transformer Temperature) datasets, publicly available in (Wu et al., 2021). These datasets contain measurements collected from electricity transformers, where the target variable is the oil temperature, with hourly granularity for ETTh1 and ETTh2 and 15-minute granularity for ETTm1 and ETTm2. The results reported in Tab. 18 show that exogenous variables can have substantially different effects on forecasting performance depending on both the forecasting horizon and the specific covariates employed. This behavior may reflect current limitations of foundation models in modeling the relationship between covariates and target variables in zero-shot settings, which remains an active research challenge. At the same time, these experiments show that the benchmarking considerations discussed throughout this work also apply to foundation models.

*Table 18.* Comparison (MSE, MAE) of Chronos2 with and without covariates for multiple windows and horizon 96. Best average results are in bold.

| Dataset | Window | w/ exog. | | w/out exog. | |
|---|---|---|---|---|---|
| | | MSE | MAE | MSE | MAE |
| Solar | 96 | $1.029 \pm$ .000 | $0.501 \pm$ .000 | $\mathbf{0.771} \pm$ .000 | $\mathbf{0.350} \pm$ .000 |
| | 336 | $0.186 \pm$ .000 | $0.183 \pm$ .000 | $\mathbf{0.167} \pm$ .000 | $\mathbf{0.176} \pm$ .000 |
| | 720 | $0.159 \pm$ .000 | $0.170 \pm$ .000 | $\mathbf{0.153} \pm$ .000 | $\mathbf{0.168} \pm$ .000 |
| Traffic | 96 | $0.726 \pm$ .000 | $0.369 \pm$ .000 | $\mathbf{0.564} \pm$ .000 | $\mathbf{0.301} \pm$ .000 |
| | 336 | $0.381 \pm$ .000 | $0.231 \pm$ .000 | $\mathbf{0.370} \pm$ .000 | $\mathbf{0.222} \pm$ .000 |
| | 720 | $0.359 \pm$ .000 | $0.222 \pm$ .000 | $\mathbf{0.351} \pm$ .000 | $\mathbf{0.215} \pm$ .000 |
| Weather | 96 | $0.326 \pm$ .000 | $0.261 \pm$ .000 | $\mathbf{0.288} \pm$ .000 | $\mathbf{0.238} \pm$ .000 |
| | 336 | $\mathbf{0.171} \pm$ .000 | $\mathbf{0.191} \pm$ .000 | $0.179 \pm$ .000 | $0.197 \pm$ .000 |
| | 720 | $\mathbf{0.154} \pm$ .000 | $\mathbf{0.181} \pm$ .000 | $0.159 \pm$ .000 | $0.186 \pm$ .000 |
| ETTm1 | 96 | $5.667 \pm$ .000 | $0.961 \pm$ .000 | $\mathbf{0.919} \pm$ .000 | $\mathbf{0.550} \pm$ .000 |
| | 336 | $0.354 \pm$ .000 | $0.345 \pm$ .000 | $\mathbf{0.346} \pm$ .000 | $\mathbf{0.333} \pm$ .000 |
| | 720 | $\mathbf{0.314} \pm$ .000 | $\mathbf{0.323} \pm$ .000 | $0.342 \pm$ .000 | $0.332 \pm$ .000 |
| ETTm2 | 96 | $0.436 \pm$ .000 | $0.380 \pm$ .000 | $\mathbf{0.246} \pm$ .000 | $\mathbf{0.301} \pm$ .000 |
| | 336 | $\mathbf{0.183} \pm$ .000 | $\mathbf{0.251} \pm$ .000 | $0.193 \pm$ .000 | $0.252 \pm$ .000 |
| | 720 | $0.181 \pm$ .000 | $0.245 \pm$ .000 | $\mathbf{0.176} \pm$ .000 | $\mathbf{0.237} \pm$ .000 |
| ETTh1 | 96 | $\mathbf{0.462} \pm$ .000 | $0.416 \pm$ .000 | $0.468 \pm$ .000 | $\mathbf{0.414} \pm$ .000 |
| | 336 | $\mathbf{0.398} \pm$ .000 | $\mathbf{0.382} \pm$ .000 | $0.423 \pm$ .000 | $0.389 \pm$ .000 |
| | 720 | $\mathbf{0.370} \pm$ .000 | $0.375 \pm$ .000 | $0.378 \pm$ .000 | $\mathbf{0.373} \pm$ .000 |
| ETTh2 | 96 | $0.360 \pm$ .000 | $0.361 \pm$ .000 | $\mathbf{0.340} \pm$ .000 | $\mathbf{0.352} \pm$ .000 |
| | 336 | $0.326 \pm$ .000 | $0.351 \pm$ .000 | $\mathbf{0.308} \pm$ .000 | $\mathbf{0.330} \pm$ .000 |
| | 720 | $\mathbf{0.300} \pm$ .000 | $0.333 \pm$ .000 | $0.311 \pm$ .000 | $\mathbf{0.331} \pm$ .000 |

