# OpenReview forum: "Position: Current Benchmarking Hinders Real Progress in Deep Learning for Time Series Forecasting"
_ICML.cc/2026/Position_Paper_Track — ICML 2026 Position Paper Track regular_

### Official Review · Reviewer_m8yE · 2026-02-27

**Significance:** 1
**Argument Clarity:** 2
**Rating:** 2
**Confidence:** 5

**Questions:**

Overall, I think this paper touches on a very interesting topic, and my main concern is whether studying such a restrictive set of datasets correlates with a broader behavior of the models observed on large-scale benchmarks such as GIFT-Eval.
My questions below (and weaknesses) reflect this interrogation.

1.	What is the reason for not using a broader benchmark for this study? In GIFT-Eval, many models have reproducible notebooks and it would be great if this paper could provide a complementary point of view on those for supervised methods they study.
2.	Spatial, temporal, and exogenous variable aspects of the different models are often ablated in TSFMs. Would this warrant a discussion in this paper too, and is it different from supervised methods? For instance, the ranking flips (if any) between fev-bench and GIFT-Eval provide a pretty solid ground for understanding the importance of exogenous variables and native multivariate support.

**Alternative Views Section:**

Yes

**Compliance With Llm Reviewing Policy A Conservative:**

Affirmed.

**Discussion Potential:**

2

**Final Justification:**

I decided to maintain my score.

While I really like the idea of this paper, which seeks to carefully disentangle the importance of the different components in deep forecasting methods, I believe it falls short of providing substantial evidence due to its rather restricted benchmarking. The authors push back on this, stating that their goal has never been to compare the models in terms of performance, but then I do not see why all the key takeaways are drawn from tables comparing the different models.

If the goal is to merely show benchmarking the importance of component X in that it changes the performance of the considered methods, then my claim still holds: this change should be evaluated on a very broad set of datasets, across different seeds and, eventually, with statistical tests of significance. Otherwise, one can easily confuse the noise of the measured performance with the signal, suggesting a genuine difference, and provide any constructive suggestions for the design of a more careful benchmarking pipeline.

**Paper Summary:**

This position paper pinpoints several drawbacks of the evaluation of current time series forecasting methods. The first one is the model configuration that is related to how the model deals with multiple time series. Second is to ablate the pre- and post-processing and incorporation of exogenous variables into the different models. Third is to consider whether a given model processes the time correlations within a given time series. Finally, the fourth one is to ablate the impact of the spatial processing on the model performance. The authors’ main point is to say that altering any of these design choices can have a significant impact on the model’s performance, yet they are often obscured in the published papers and not ablating properly. This lack of careful disentanglement between the impact of these design choices leads to a lack of understanding of what drives the progress in the field further. It also hinders establishing the best practices in the field.

**Position:**

Yes

**Position In Title:**

Yes

**Related Work:**

1

**Strengths And Weaknesses:**

*Strengths*

1.	Careful exposition of several important design choices in time series forecasting methods
2.	Clear presentation of the main findings
3.	Experimental results are interesting and insightful

*Weaknesses*

1.	Unfortunately, this paper completely overlooks the most recent benchmarks, such as GIFT-Eval and fev-bench. Most of the findings presented in this paper are derived only from the common long-term time series datasets such as Electricity, Weather, Traffic, and Solar. A recent work [1] showed that a shallow one-head transformer can beat TSFMs (and also PatchTST) on them. But in fact, this is of course not representative of the true power of the compared models, but rather indicative of the peculiarities of these datasets.
2.	Similarly to the point above, it is not clear why methods such as Crossformer were included in the study. On GIFT-Eval, this method is the worst across all methods considered, including AutoArima, Naïve, and Seasonal Naive. In my opinion, any generalized findings like the ones presented here should be carried out using methods that were evaluated on a broader benchmark on which they perform well.

[1] SAMformer: Unlocking the Potential of Transformers in Time Series Forecasting with Sharpness-Aware Minimization and Channel-Wise Attention, ICML’24

**Support:**

2

---

> ### Author Rebuttal · Authors · 2026-03-31
>
> Thank you for considering our work and findings clear, and our experiment results insightful. Though we can see where your concerns come from, we do not agree and believe there might have been some misunderstandings. We clarify below.
>
> >(Weakness 1 & question 1) This paper completely overlooks the most recent benchmarks, such as GIFT-Eval and fev-bench … What is the reason for not using a broader benchmark for this study?
>
> There might be a misunderstanding here. The reason we did not use a larger benchmark for this study is quite simple: this is not a benchmarking paper. Indeed, our objective has never been to benchmark current architectures, but rather to discuss our belief that the current dominant approach to benchmarking them is flawed. To do this, we used datasets that are commonly used in the literature and that have often been used to propose new architectures. We agree that this can be limited to show which architecture outperforms the others (indeed, we show that simple baselines match the performance of current methods), but this is not the paper's purpose. We believe that our empirical results perfectly show why our position is relevant. Indeed, from your review, you seem to agree that the points we raise are a relevant concern, and the position is well laid out. Note that having empirical results to support one’s position is not even a requirement in position papers. Finally, we acknowledge that someone might believe that the only issue with current benchmarking is in the quality of the datasets being used, and this is already discussed as an alternative view in Section 5. We will expand the discussion in this direction to better highlight the growing effort in producing better benchmarks.
>
> >(Weakness 2) It is not clear why methods such as Crossformer were included in the study. On GIFT-Eval, this method is the worst across all methods considered, including AutoArima, Naïve, and Seasonal Naive. In my opinion, any generalized findings like the ones presented here should be carried out using methods that were evaluated on a broader benchmark on which they perform well.
>
> Here, our answer is similar to that in the previous point. Since our objective has never been to provide a ranking of existing state-of-the-art methods, we do not see any issue in including a widely used method (the ICLR paper has 2000+ citations) that has been shown not to perform as competitively as expected. Conversely, as the goal of our paper is to identify the issues that may have led to such misleading conclusions, its inclusion in the discussion is fully justified.
>
> >(Question 2) Spatial, temporal, and exogenous variable aspects of the different models are often ablated in TSFMs. Would this warrant a discussion in this paper too, and is it different from supervised methods? For instance, the ranking flips (if any) between fev-bench and GIFT-Eval provide a pretty solid ground for understanding the importance of exogenous variables and native multivariate support.
>
> This is an interesting point, but again, as our goal is not to draw definitive answers on which component works best, such a thorough analysis is out of scope for this position paper. However, we will mention that future work (in particular, a benchmarking/evaluation paper) ought to carry out a more comprehensive evaluation of this aspect in appropriate benchmarks.
>
> To summarize our answers, we believe that the role of a position paper like this is to raise concerns and awareness regarding some of the issues present in our field. Extensive and advanced evaluation on large-scale benchmarks, though already discussed in our paper as a next step, is out of scope here. We are also a bit confused by the score for significance (Significance=1), as the review mentions that our position is ‘very interesting’ and that results are  ‘interesting and insightful’. We hope to have clarified the raised concerns and are happy to continue the discussion if needed.

---

> > ### Author Rebuttal · Reviewer_m8yE · 2026-04-02
> >
> > I thank the authors for their reply. Unfortunately, my concerns remain.
> >
> > I argue against building a position on something that may not reflect the actual progress in the field. I think that evaluating on 4 datasets may not be enough to draw any conclusion from it. Similarly, using a method (despite its 2000+ citations) that ranks last on a commonly used large-scale study representative, to some extent, of the forecasting task, may not inform the same conclusions as an analysis of well-performing deep learning methods.
> >
> > I will maintain my score.

---

### Official Review · Reviewer_u3kw · 2026-03-11

**Significance:** 4
**Argument Clarity:** 3
**Rating:** 5
**Confidence:** 4

**Questions:**

- Q1: Have the experiments in Sec. 3 been aggregated across multiple runs (with different random seeds)?
- Q2: What about the Forecasting Model Card would need to change to be appropriate for Time Series Foundation Models (mentioned as a limitation)?

**Alternative Views Section:**

Yes

**Compliance With Llm Reviewing Policy A Conservative:**

Affirmed.

**Discussion Potential:**

3

**Final Justification:**

I agree with the other reviewers that the lack of treatment of foundation models is a substantial weakness of the work. Also, while other datasets (newer large-scale benchmarks) could have been used to strengthen the analysis, I side with the authors in saying they are not necessary to defend the core position. These weaknesses do not outweigh the merits of this work.

My concerns have been adequately addressed. I raised my score to a full Accept (from borderline acceptance).

**Paper Summary:**

The work analyzes current practices in reporting findings on deep learning models for time series forecasting. By assessing existing work and conducting new experiments, it is shown that moderate changes to the experimental setting can have substantial ramifications for model performance. The call to action is therefore to (a) more carefully design experiments to ensure apples-to-apples comparisons and (b) to be transparent about the details of the model and evaluation scenario.

**Position:**

Yes

**Position In Title:**

Yes

**Related Work:**

2

**Strengths And Weaknesses:**

**Strengths:**
- The work is well-written.
- The motivation for why the current practices are hindering progress is very convincing.
- The experiments in Sec. 3.1/3.2/3.3/3.4 adequately show the issues with current practices.
- The call to action is convincing. Its first part is essentially a call for good experimental design. Its second part, the Forecasting Model Card, is easy to follow as it only makes practices transparent without necessitating large behavioural changes.

**Weaknesses:**
- There is an alternate view that is not properly discussed (in both Sec. 1 and 5): Say, a work proposes a new architecture with novel components X and Y, and the overall system is better than the previous SOTA. Then, oftentimes, the conclusion from being better than all baselines is that X and Y are superior components (for the tasks investigated). However, this usually does not follow logically without appropriate ablation studies. The alternative view would be: The lack of ablation studies and similar component-comparison experiments is the real issue (experiments not suitable to support the logical conclusions), not bad reporting. I believe this view is not adequately discussed in the work (e.g., also in Sec. 4, first call).
- As an additional item in the call to action (Sec. 4), the need for appropriate statistical tests should be named. More often than not, loose/subjective statements like "the rank is much better" or "the mean error is substantially lower" are used to substantiate conclusions about what components/models are superior. Instead, statistical tests with a set significance level should be used for any key conclusions of a publication, wherever possible.
- The stability of the results across multiple random seeds (e.g., different model initializations, different data sampling, etc.) is not adequately discussed. This is important for (a) the experiments carried out in Sec. 3 and (b) the suggested forecasting model card. If computationally feasible, any statements about model quality should be based on aggregated metrics across multiple such runs (e.g., by stating the average and standard deviation).
#### Minor Comments
- Section 2.1 could also define "transductive" and "inductive" settings, as not all readers might be familiar with them.
- Fig. 1 (a) legend error: The size of the circles is disconnected from the legend sizes (or the labels are wrong). The brown circle for TimeMixer is larger than the grey 5.00 GB circle in the legend, but labeled as less (2.25GB).
- Friendly note: Brigato et al. (2025) has since been published in TMLR.
- The key insights from Tab. 1/2 (etc.) might be easier to grasp if the highlights were not row-wise but per-detaset column-wise, showing that the best model changes depending on the configurations being contrasted. Seeing how ranks would change would be an alternative.

**Support:**

3

---

> ### Author Rebuttal · Authors · 2026-03-31
>
> We appreciate that the reviewer finds the motivation of our work and the call to action convincing, our experiments adequate to support our position, and the model cards a practical solution.
> >(Weakness 1) There is an alternate view that is not properly discussed (in both Sec. 1 and 5). ... The alternative view would be: The lack of ablation studies and similar component-comparison experiments is the real issue (experiments not suitable to support the logical conclusions), not bad reporting. I believe this view is not adequately discussed in the work (e.g., also in Sec. 4, first call).
>
> This is a good point; however, it is not entirely alternative to our position as it is actually part of what we discuss as a problem. Note that the issues we discuss are not limited to reporting. The lack of a good ablation study is often a consequence of overlooking important design aspects that instead have an impact on performance. Key differences among the architectures being compared are often not adequately discussed, and as such, not included in ablation studies (if any).  Indeed, good ablation studies are a way to make these differences explicit, but the discussion should go beyond simply showing how they impact the architecture being proposed. Thanks for the comment; we will include this aspect in the discussion and update the call to action to clarify this aspect.
> >(Weakness 2) As an additional item in the call to action (Sec. 4), the need for appropriate statistical tests should be named.
>
> We fully agree. We will mention this in the call to action.
> >(Weakness 3 & Question1) The stability of the results across multiple random seeds (e.g., different model initializations, different data sampling, etc.) is not adequately discussed. This is important for (a) the experiments carried out in Sec. 3 and (b) the suggested forecasting model card. ... Have the experiments in Sec. 3 been aggregated across multiple runs (with different random seeds)?
>
> All the results, shown both in the main body and in the appendix, report the standard deviations across at least 3 runs with different random seeds. We will make this more explicit in the revised version and include a comment on the stability of results, which we observe to have low variance across runs. Moreover, we agree on the importance of considering the randomness of the experiments when evaluating model designs, and we will underline it in the call to action. Thank you for pointing this out.
>
> >(Question2) What about the Forecasting Model Card would need to change to be appropriate for Time Series Foundation Models (mentioned as a limitation)?
>
> This is an interesting discussion. The forecasting model cards should be extended with an additional field to include information on the datasets used for training (e.g., whether they were trained on synthetic datasets [1] and so on). Moreover, the model configuration field (D1) would not be strictly needed, as Time Series Foundation Models are global by default.
> > (Minor comments) Section 2.1 could also define "transductive" and "inductive" settings, legend error in the plot, publication update on Brigato et al. (2025) and suggestion for table update.
>
>
> Thank you for pointing this out. We will update the paper accordingly.
>
> We sincerely thank you for the review; expanding the discussion to cover these points will definitely improve the paper. We hope we have addressed all of your questions.
>
> .[1] N. Boris et al., Zero-shot Forecasting by Simulation Alone, ICLR 2026

---

> > ### Author Rebuttal · Reviewer_u3kw · 2026-04-05
> >
> > I want to thank the authors for their response. My concerns have been addressed. I raised my score to a full Accept.
> >
> > I agree with the other reviewers that the lack of treatment of foundation models is a substantial weakness of the work. Also, while other datasets (newer large-scale benchmarks) could have been used to strengthen the analysis, I side with the authors in saying they are not necessary to defend the core position. These weaknesses do not outweigh the merits of this work.

---

### Official Review · Reviewer_mGZF · 2026-03-13

**Significance:** 3
**Argument Clarity:** 2
**Rating:** 4
**Confidence:** 4

**Questions:**

1. In section 3.3, when stating that the results match SOTA is this drawing metrics from these new comparisons (fair or not) or from existing publications? Would these different 'fair' comparisons be lowering performance on some models, even if they are now more explainable?
2. Keeping everything fixed and the same across models while varying one design component aligns with the traditional scientific method, but ignores complex relationships between components (as covariates). How would this framework address that?

**Alternative Views Section:**

Yes

**Compliance With Llm Reviewing Policy A Conservative:**

Affirmed.

**Discussion Potential:**

2

**Final Justification:**

A couple rounds of feedback with the authors during the review process resulted in most of my questions being answered. I think experiments are not necessary but can highlight the position in this paper. I raised my score by 1.

**Paper Summary:**

This papers argues that the fundamental problem in current benchmarking practices is that they do not go far enough into identifying the factors responsible for performance differences in time series forecasting.

**Position:**

Yes

**Position In Title:**

Yes

**Related Work:**

2

**Strengths And Weaknesses:**

- The position is important and fairly clear in scope.
    - The text tends to have long run-on sentences that detract from the overall readability of the work.
    - I would not expect a position paper to provide new comparisons of models on benchmarks; this feels more appropriate for a research paper.
    - The categorization of design dimensions is well-motivated and explained, with support from the existing literature.
    - The proposed forecasting model card seems like a list of parameters and design components rather than something more useful like a proposed procedure or pipeline for evaluation.

**Support:**

3

---

> ### Author Rebuttal · Authors · 2026-03-31
>
> We sincerely appreciate that you consider our position important and clear, as well as the explanation and categorization of design dimensions. We provide answers below to the questions and concerns.
>
>
> >(Weakness 1) I would not expect a position paper to provide new comparisons of models on benchmarks; this feels more appropriate for a research paper.
>
> The experiments are not meant to do an extensive benchmarking of existing models, but rather to provide evidence to support our position and show where it comes from. We will clarify this in the revision.
>
>
> >(Weakness 2) The proposed forecasting model card seems like a list of parameters and design components rather than something more useful like a proposed procedure or pipeline for evaluation.
>
> The aim of the model card is not to define a pipeline for evaluation, and clearly, it cannot solve the current issues in benchmarking on its own. The required steps are discussed in depth in the call to action, and their full implementation is outside the scope of a position paper. Nonetheless, we include model cards and the associated template as a first concrete step towards a sound model comparison, as they provide a simple way to shed light on what really matters in model architectures in a neat and standardized format. This is particularly useful since many important design choices are often overlooked and need to be extracted from code or supplementary materials.
>
> >(Question 1) In section 3.3, when stating that the results match SOTA is this drawing metrics from these new comparisons (fair or not) or from existing publications? Would these different 'fair' comparisons be lowering performance on some models, even if they are now more explainable?
>
> Results in Section 3.3 are obtained by relying on the original implementation of each method from the respective papers and adding covariates in the way that best suits each architecture. Indeed, the results are coherent with what one would expect from these models on the selected datasets (even slightly better in some cases). Results in 3.3 show that SOTA methods perform close to simpler baselines when key designs are taken into account. All the experiments were run in the same evaluation setup.
>
> >(Question 2) Keeping everything fixed and the same across models while varying one design component aligns with the traditional scientific method, but ignores complex relationships between components (as covariates). How would this framework address that?
>
> We do not claim that models should be simplified at the expense of performance for fairer comparisons. Rather, when attributing performance gains to specific components, one should account for the differences in other design dimensions between models being compared. Indeed, design dimensions sometimes cannot be disentangled. In those cases, we call for explicitly acknowledging those differences, rather than hiding them, to avoid misleading conclusions. This topic is addressed in the second alternative view, and we will expand that discussion in the revised version to clarify how our call to action applies when disentanglement is not possible.
>
>
> Thank you for the interesting point raised in this discussion. We hope to have addressed your concerns. Should there be any remaining questions or points needing clarification, we would be happy to continue the discussion.

---

> > ### Author Rebuttal · Reviewer_mGZF · 2026-04-02
> >
> > 1. New comparisons: Can you clarify this here instead of just in revision?
> >
> > 2. Model cards: To clarify, the card is more of a set of evaluation criteria? To help guide selection of action steps later?
> >
> > 3. "adding covariates in the way that best suits each architecture" could you be more detailed please?
> >
> > 4. How do you think your "call to action applies when disentanglement is not possible"?

---

### Official Review · Reviewer_89ou · 2026-03-13

**Significance:** 2
**Argument Clarity:** 3
**Rating:** 3
**Confidence:** 4

**Questions:**

1. There is a typo in Eq. (1). $x_{t:t+H}^i$ should be $x_{t+1:t+H}^i$. Also there are two $\approx$.
2. Is it possible to provide the source for review?

**Alternative Views Section:**

Yes

**Compliance With Llm Reviewing Policy A Conservative:**

Affirmed.

**Discussion Potential:**

3

**Paper Summary:**

This paper addresses the benchmarking of time series forecasting model. It argues that differences in crucial design dimensions are overlooked when comparing architecture, leading to inconsistent outcomes. It proposes to consider four key components in benchmarking, including model configuration, preprocessing and exogenous variables, temporal processing, and spatial processing. As a concrete step, it proposes an auxiliary forecasting model card, i.e., a template with a set of fields to characterize existing and new forecasting architectures based on key design choices.

**Position:**

Yes

**Position In Title:**

Yes

**Related Work:**

3

**Strengths And Weaknesses:**

Strengths:
1. It addresses an important problem in the time series community.
2. The discussion is clear and actionable. It considers four key components in deep learning models for time series forecasting, including model configuration, preprocessing and exogenous variables, temporal processing, and spatial processing. Here the spatial processing means the processing of inter-series dependencies in modeling.
3. It proposes a concrete step, the auxiliary forecasting model card, a template with a set of fields to characterize existing and new forecasting architectures based on key design choices.

Weaknesses:
1. Limited discussion of the time series foundation forecasting model, e.g., Chronos-2. It is suggested to include Chronos-2 in the empirical studies.
2. The four datasets tested are too limited and may be not representative.  The empirical studies look oversimplified.
3. The scope of this paper is a little limited.

**Support:**

3

---

> ### Author Rebuttal · Authors · 2026-03-31
>
> Thank you for recognizing the importance and clarity of our work and appreciating the forecasting model cards.
>
> > (Weakness 1) Limited discussion of the time series foundation forecasting model
>
>
> The issues of misattribution of performance due to benchmarking apply directly to foundation models too, as the same benchmarking problems may mislead the design of foundation models as well.
> For the empirical evaluation, we focus on models that allow multiple training under different configurations. Doing a similar analysis for foundation models would require a completely different setup. Indeed, they are designed for broad, cross-domain applicability and are typically provided as ready-to-use. Moreover, retraining foundation models across multiple configurations would be computationally prohibitive and would not change the conclusions of our discussions. We believe our position remains relevant, with the presented experiments already supporting it. We appreciate the suggestion and will clarify this aspect in the revised version.
> Let us know if anything requires more discussion here.
>
>
> > (Weakness 2 & 3)  The four datasets tested are too limited and may be not representative. The empirical studies look oversimplified. The scope of this paper is a little limited.
>
> We selected these datasets as they are widely used and are often associated with the flawed benchmarking practices discussed in the paper.
> We agree that they are neither the most challenging nor the most interesting. Indeed, our experiments confirm this since very simple models often have competitive performances here.
> However, the objective of our position paper is not to benchmark current methods, but rather to show shortcomings in current benchmarking practices. This is why we believe choosing the same datasets used in literature is appropriate.  Our discussion and experiments are designed to support this point, not to establish which method outperforms others. We acknowledged the importance of datasets in benchmarking, and we addressed it in the call to action. Moreover, we discuss some of these aspects also in the first alternative view in Section 5.
>
> >(Question1) There is a typo in Eq. (1)
>
> Thank you for pointing out the typo, we will fix it in the updated version.
>
> >(Question2) Is it possible to provide the source for review?
>
> Please find the code to reproduce experiments here: https://anonymous.4open.science/r/position_benchmarking-81C4
>
>
> We hope that the clarifications above address all of your concerns.

---

> > ### Author Rebuttal · Reviewer_89ou · 2026-04-03
> >
> > Thanks for the rebuttal and providing the source codes.
> > Generally the discussion on the time series foundation model sounds reasonable, but some experiments, even simplified experiments, would make the argument more convincing.
> >
> > At this moment, I would like to maintain my score.

---

### Decision · Program_Chairs · 2026-04-30

**Decision:**

Accept (regular)

**Comment:**

This position paper addresses a significant and timely issue in the time series forecasting community: the flawed nature of current benchmarking and evaluation practices. The authors propose a concrete, actionable solution in the form of Forecasting Model Cards to ensure transparent and fair comparisons across different model architectures.

The submission generated mixed but generally supportive reviews.
Strengths:

The motivation is highly convincing, and the problem it addresses is widely recognized within the community.
The proposed solution (Forecasting Model Cards) is practical and actionable without requiring massive behavioral shifts from researchers.
The authors provided a strong, responsive rebuttal, agreeing to add discussions on statistical significance, the necessity of rigorous ablation studies, and the limitations of current datasets.

Weaknesses and Reviewer Disagreements:

A major point of contention was the empirical setup. Reviewers pointed out that the paper relied on only four traditional datasets and omitted recent large-scale benchmarks (like GIFT-Eval) and Time Series Foundation Models (TSFMs).
One reviewer heavily penalized the paper for this, arguing that the limited datasets and the inclusion of underperforming models (like Crossformer) undermined the findings.
However, the authors argued (and the majority of reviewers ultimately agreed) that this is a position paper, not a benchmarking paper. The experiments provided are illustrative—designed to prove that current evaluation practices are flawed and easily manipulated, not to establish a new state-of-the-art ranking.

While the empirical evidence is somewhat narrow, it serves its intended purpose of supporting the paper's core thesis. The authors' rebuttal clarified the scope of their claims and their commitment to expanding the discussion on TSFMs and ablation studies. Overall, the paper advocates for a needed methodological shift in the community and presents a compelling, actionable framework to achieve it.